# SHAPLEY EXPLANATION NETWORKS

**Rui Wang    Xiaoqian Wang    David I. Inouye**
School of Electrical and Computer Engineering
Purdue University
West Lafayette, IN 47906
{ruiw, joywang, dinouye}@purdue.edu

## ABSTRACT

Shapley values have become one of the most popular feature attribution explanation methods. However, most prior work has focused on post-hoc Shapley explanations, which can be computationally demanding due to its exponential time complexity and preclude model regularization based on Shapley explanations during training. Thus, we propose to incorporate Shapley values themselves as latent representations in deep models—thereby making Shapley explanations first-class citizens in the modeling paradigm. This intrinsic explanation approach enables layer-wise explanations, explanation regularization of the model during training, and fast explanation computation at test time. We define the *Shapley transform* that transforms the input into a *Shapley representation* given a specific function. We operationalize the Shapley transform as a neural network module and construct both shallow and deep networks, called SHAPNETS, by composing Shapley modules. We prove that our Shallow SHAPNETS compute the exact Shapley values and our Deep SHAPNETS maintain the missingness and accuracy properties of Shapley values. We demonstrate on synthetic and real-world datasets that our SHAPNETS enable layer-wise Shapley explanations, novel Shapley regularizations during training, and fast computation while maintaining reasonable performance. Code is available at https://github.com/inouye-lab/ShapleyExplanationNetworks.

## 1 INTRODUCTION

Explaining the predictions of machine learning models has become increasingly important for many crucial applications such as healthcare, recidivism prediction, or loan assessment. Explanations based on feature importance are one key approach to explaining a model prediction. More specifically, additive feature importance explanations have become popular, and in Lundberg & Lee (2017), the authors argue for theoretically-grounded additive explanation method called SHAP based on Shapley values—a way to assign credit to members of a group developed in cooperative game theory (Shapley, 1953). Lundberg & Lee (2017) defined three intuitive theoretical properties called local accuracy, missingness, and consistency, and proved that only SHAP explanations satisfy all three properties.

Despite these elegant theoretically-grounded properties, exact Shapley value computation has exponential time complexity in the general case. To alleviate the computational issue, several methods have been proposed to approximate Shapley values via sampling (Strumbelj & Kononenko, 2010) and weighted regression (Kernel SHAP), a modified backpropagation step (Deep SHAP) (Lundberg & Lee, 2017), utilization of the expectation of summations (Ancona et al., 2019), or making assumptions on underlying data structures (Chen et al., 2019). To avoid approximation, the model class could be restricted to allow for simpler computation. Along this line, Lundberg et al. (2020) propose a method for computing exact Shapley values for tree-based models such as random forests or gradient boosted trees. However, even if this drawback is overcome, prior Shapley work has focused on *post-hoc* explanations, and thus, the explanation approach cannot aid in model design or training.

On the other hand, Generalized Additive Models (GAM) as explored in Lou et al. (2012; 2013); Caruana et al. (2015) (via tree boosting), Chen et al. (2017a) (via kernel methods), and Wang et al. (2018); Agarwal et al. (2020) (via neural networks) can be seen as interpretable model class that exposes the exact Shapley explanation directly. In particular, the output of a GAM model is simply the summation of interaction-free functions: $f_{\mathrm{GAM}}(\boldsymbol{x}) = \sum_s f_s(x_s)$, where $f_s(\cdot)$ are univariate functions

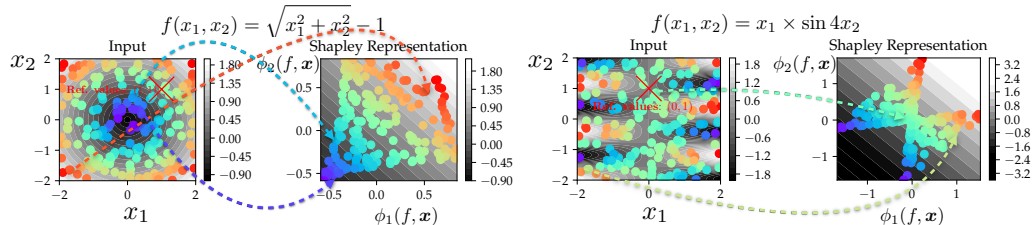

Figure 1: Shapley representations span beyond one-dimensional manifolds and depend on both the inner function and the reference values. In both groups, the gray-scale background indicates the respective function value while the rainbow-scale color indicates correspondence between input (left) and Shapley representation (right) along with function values—red means highest and purple means lowest function values. The red cross represents the reference values. More details in subsection 2.1.

that can be arbitrarily complex. Interestingly, the Shapley explanation values, often denoted $\phi_s(\boldsymbol{x}, f)$, for a GAM are exactly the values of the independent function, i.e., $\forall s, \phi_s(\boldsymbol{x}, f_{\text{GAM}}) = f_s(x_s)$. Hence, the prediction and the corresponding exact SHAP explanation can be computed simultaneously for GAM models. However, GAM models are inherently limited in their representational power, particularly for perceptual data such as images in which deep networks are state-of-the-art. Thus, prior work is either post-hoc (which precludes leveraging the method during training) or limited in its representational power (e.g., GAMs).

To overcome these drawbacks, we propose to incorporate Shapley values themselves as learned latent representations (as opposed to post-hoc) in deep models—thereby making Shapley explanations first-class citizens in the modeling paradigm. Intuitive illustrations of such representation are provided in Fig. 1 and detailed discussion in subsection 2.1. We summarize our core contributions as follows:

- We formally define the *Shapley transform* and prove a simple but useful linearity property for constructing networks.

- We develop a novel network architecture, the SHAPNETs, that includes Shallow and Deep SHAPNETs and that intrinsically provides layer-wise explanations (i.e., explanations at every layer of the network) in the same forward pass as the prediction.

- We prove that Shallow SHAPNET explanations are the *exact* Shapley values—thus satisfying all three SHAP properties—and prove that Deep SHAPNET explanations maintain the missingness and local accuracy properties.

- To reduce computation, we propose an instance-specific dynamic pruning method for Deep SHAPNETs that can skip unnecessary computation.

- We enable explanation regularization based on *Shapley values* during training because the explanation is a latent representation in our model.

- We demonstrate empirically that our SHAPNETs can provide these new capabilities while maintaining comparable performance to other deep models.

**Dedicated Related Works Section** We present related works above and in the text where appropriate. Due to space limit, we refer to Appendix D for a dedicated literature review section.

## 2 SHAPLEY EXPLANATION NETWORKS

**Background** We give a short introduction to SHAP explanations and their properties as originally introduced in Lundberg & Lee (2017). Given a model $f : \mathbb{R}^d \mapsto \mathbb{R}$ that is not inherently interpretable (e.g., neural nets), additive feature-attribution methods form a linear approximation of the function over simplified binary inputs, denoted $\boldsymbol{z} \in \{0, 1\}^d$, indicating the "presence" and "absence" of each feature, respectively: i.e., a local linear approximation $\eta(\boldsymbol{z}) = a_0 + \sum_{i=1}^d a_i z_i$. While there are different ways to model "absence" and "presence", in this work, we take a simplified viewpoint: "presence" means that we keep the original value whereas "absence" means we replace the original value with a reference value, which has been validated in Sundararajan & Najmi (2020) as Baseline Shapley. If we denote the reference vector for all features by $\boldsymbol{r}$, then we can define a simple mapping

function between $z$ and $x$ as $\Psi_{x,r}(z) = z \odot x + (1 - z) \odot r$, where $\odot$ denotes element-wise product (eg, $\Psi_{x,r}([0, 1, 0, 1, 1]) = [r_1, x_2, r_3, x_4, x_5]$). A simple generalization is to group certain features together and consider including or removing all features in the group. Lundberg & Lee (2017) propose three properties that additive feature attribution methods should intuitively satisfy. The first property called *local accuracy* states that the approximate model $\eta$ at $z = \mathbf{1}$ should match the output of the model $f$ at the corresponding $x$, i.e., $f(x) = \eta(\mathbf{1}) = \sum_{i=0}^{d} a_i$. The second property called *missingness* formalizes the idea that features that are "missing" from the input $x$ (or correspondingly the zeros in $z$) should have zero attributed effect on the output of the approximation, i.e., $z_i = 0 \Rightarrow a_i = 0$. Finally, the third property called *consistency* formalizes the idea that if one model always sees a larger effect when removing a feature, the attribution value should be larger (see (Lundberg & Lee, 2017) for full definitions of the three properties and their discussions).

**Definition 1** (SHAP Values (Lundberg & Lee, 2017)). *SHAP values are defined as:*

$$\phi_i(f, x) \triangleq \frac{1}{d} \sum_{z \in \mathcal{Z}_i} \binom{d-1}{\|z\|_1}^{-1} \left[ f\left(\Psi_{x,r}(z \cup \{i\})\right) - f\left(\Psi_{x,r}(z)\right) \right], \tag{1}$$

*where $\|z\|_1$ is the $\ell_1$ norm of $z$ given a set $\mathcal{Z}_i = \{z \in \{0, 1\}^d : z_i = 0\}$ and $z \cup \{i\}$ represents setting the $i$-th element in $z$ to be 1 instead of 0.*

### 2.1 SHAPLEY TRANSFORM AND SHAPLEY REPRESENTATION

For the sake of generality, we will define the Shapley transform in terms of tensors (or multi-dimensional arrays). Below in Def. 2 we make a distinction between tensor dimensions that need to be explained and other tensor dimensions. For example, in image classification, usually an explanation (or attribution) value is required for each pixel (i.e., the spatial dimensions) but all channels are grouped together (i.e., an attribution is provided for each pixel but not for each channel of each pixel). We generalize this idea in the following definitions of explainable and channel dimensions.

**Definition 2** (Explainable Dimensions and Channel Dimensions). *Given a tensor representation $x \in \mathbb{R}^{D \times C} \equiv \mathbb{R}^{(d_1 \times d_2 \times \cdots) \times (c_1 \times c_2 \times \cdots)}$, the tensor dimensions $D \equiv d_1 \times d_2 \times \cdots$ that require attribution will be called* explainable dimensions *and the other tensor dimensions $C \equiv c_1 \times c_2 \times \cdots$ will be called the* channel dimensions.

As a simplest example, if the input is a vector $x \in \mathbb{R}^{d \times 1}$, then $D = d$ and $C = 1$, i.e., we have one (since $C = 1$) importance value assigned to each feature. For images in the space $\mathbb{R}^{h \times w \times c}$ where $h, w$ denote the height and width, respectively, the explainable dimensions correspond to the spatial dimensions (i.e., $D = h \times w$) and the channel dimensions correspond to the single channel dimension (i.e., $C = c$). While the previous examples discuss tensor dimensions of the input, our Shapley transforms can also operate on latent representations (e.g., in a neural net). For example, our latent representations for image models could be in the space $\mathbb{R}^{w \times h \times c_1 \times c_2}$, where the explainable dimensions correspond to spatial dimensions (i.e., $D = h \times w$) and there are two channel dimensions (i.e., $C = c_1 \times c_2$). Given this distinction between explainable and channel dimensions, we can now define the Shapley transform and Shapley representation.

**Definition 3** (Shapley Transform). *Given an arbitrary function $f \in \mathcal{F} \colon \mathbb{R}^{D \times C} \mapsto \mathbb{R}^{C'}$ the Shapley transform $\Omega \colon \mathbb{R}^{D \times C} \times \mathcal{F} \mapsto \mathbb{R}^{D \times C'}$ is defined as:*

$$[\Omega(x, f)]_{i,j} = \phi_i(x, f_j), \quad \forall i \in \mathcal{I}_D, j \in \mathcal{I}_{C'}, \tag{2}$$

*where $\mathcal{I}_D$ denotes the set of all possible indices for values captured by the explainable dimensions $D$ (and similarly for $\mathcal{I}_{C'}$) and $f_j$ denotes the scalar function corresponding to the $j$-th output of $f$.*

**Definition 4** (Shapley Representation). *Given a function $f \in \mathcal{F}$ as in Def. 3 and a tensor $x \in \mathbb{R}^{D \times C}$, we simply define the* Shapley representation *to be: $Z \triangleq \Omega(x, f) \in \mathbb{R}^{D \times C'}$.*

Notice that in the Shapley transform, we always keep the explainable dimensions $D$ unchanged. However, the channel dimensions of the output representation space, i.e., $C'$, are determined by the co-domain of the function $f$. For example, if $f$ is a scalar function (i.e., $C' = 1$), then the attribution for the explainable dimensions is a scalar. However, if $f$ is a vector-valued function, then the attribution is a *vector* (corresponding to the vector output of $f$). A multi-class classifier is a simple example of an $f$ that is a vector-valued function (e.g., $C' = 10$ for 10-class classification tasks). In summary, the Shapley transform maintains the explainable tensor dimensions, but each explainable element of $D$ may be associated to a tensor of attribution values corresponding to each output of $f$.

## 2.2 SHALLOW SHAPNET

We now concretely instantiate our SHAPNETs by presenting our Shallow SHAPNET and prove that it produces exact Shapley values (see Lemma 1 and Theorem 2, proofs in Appendix A).

**Definition 5** (Shallow SHAPNET). *A Shallow* SHAPNET $\mathcal{G}: \mathbb{R}^{D \times C} \mapsto \mathbb{R}^{C'}$ *is defined as*

$$\mathcal{G}_f(\boldsymbol{x}) = \text{sum}^{[D]} \circ g(\boldsymbol{x}, f) = \text{sum}^{[D]} \circ \Omega(\boldsymbol{x}, f), \tag{3}$$

*where* $g(\boldsymbol{x}, f) \equiv \Omega(\boldsymbol{x}, f)$ *denotes the Shallow* SHAPNET *explanation,* $\text{sum}^{[D]}$ *denotes the summation over explainable dimensions* $D$, $f : \mathbb{R}^{D \times C} \mapsto \mathbb{R}^{C'}$ *is the underlying function, and* $C'$ *is the required output dimension for the learning task (e.g., the number of classes for classification).*

**Remark 6** (Sparsity in active sets curbs computation). By itself, this definition does not relieve the exponential computational complexity in the number of input features of computing Shapley values. Thus, to alleviate this in SHAPNETs, we promote sparsity in the computation. Consider a function $f_j$ that depends only on a subset of input features, (e.g., $f_j(x_1, x_2, x_3) = 5x_2 + x_3$ only depends on $x_2$ and $x_3$); we will refer to such a subset as the *active set*, $\mathcal{A}(f_j)$ ($\mathcal{A}(f_j) = \{2, 3\}$ for the example above). In principle, we would like to keep the cardinality of the active set low: $|\mathcal{A}(f_j)| \ll |\mathcal{I}_D|$ for $j \in \mathcal{I}_{C'}$, so that we can limit the computational overhead because features in the complementary set of $\mathcal{A}(f_j)$ (i.e., the inactive set) have trivial Shapley values of 0's for any input.

To enforce the sparsity by construction, we propose a simple but novel wrapper around an arbitrarily complex scalar function $f_j$ (which can be straightforwardly extended to vector-valued functions) called a *Shapley module* $F$ that explicitly computes the Shapley values locally but only for the *active set* of inputs (the others are zeros by construction and need not be computed) as in Fig. 2 (left).

**Definition 7** (Shapley Module). *Given a max active set size* $k \ll |\mathcal{I}_D|$ *(usually* $2 \le k \le 4$*), an arbitrarily complex function* $f : \mathbb{R}^{D \times C} \mapsto \mathbb{R}$ *parameterized by a neural network such that* $|\mathcal{A}(f)| \le k$ *by construction, and a reference vector* $\boldsymbol{r} \in \mathbb{R}^{D \times C}$, *a* Shapley module $F : \mathbb{R}^{D \times C} \mapsto \mathbb{R}^{D \times 1}$ *is defined as:* $F(\boldsymbol{x} \mid f, \boldsymbol{r}) \triangleq [\phi_1(f, \boldsymbol{x}), \dots, \phi_d(f, \boldsymbol{x})]^T$, *where* $\phi_i(f, \boldsymbol{x})$ *is computed explicitly via Def. 1 if* $x_i \in \mathcal{A}(f)$ *and* $\phi_i(f, \boldsymbol{x}) = 0$ *otherwise (by construction).*

While the sparsity of the Shapley modules significantly reduces the computational effort, only a small subset (i.e., the active set) of features affect the output of each Shapley module. Thus, to combat this issue, we consider combining the outputs of many Shapley modules that cover all the input features (i.e., $\bigcup_j \mathcal{A}(f_j) = \mathcal{I}_D$) and can even be overlapping (e.g., $\mathcal{A}(f_1) = \{1, 2\}, \mathcal{A}(f_2) = \{2, 3\}, \mathcal{A}(f_3) = \{1, 3\}$). We first prove a useful linearity property of our Shapley transforms. Without loss of generality, we can assume that the explainable dimensions and the channel dimensions have been factorized so that $D = d_1 d_2 d_3 \cdots$ and $C = c_1 c_2 c_3 \cdots$ in the following lemma.

**Lemma 1** (Linear transformations of Shapley transforms are Shapley transforms). *The linear transform, denoted by a matrix* $A \in \mathbb{R}^{C'' \times C'}$, *of a Shapley representation* $Z \triangleq \Omega(\boldsymbol{x}, f) \in \mathbb{R}^{D \times C'}$, *is itself a Shapley transform for a modified function* $\tilde{f}$:

$$A Z^T \equiv A \left( \Omega(\boldsymbol{x}, f) \right)^T = \left( \Omega \left( \boldsymbol{x}, \tilde{f} \right) \right)^T, \quad \text{where } \tilde{f}(\boldsymbol{x}) = A f(\boldsymbol{x}). \tag{4}$$

Given this linearity property and the sparsity of the output of Shapley modules (which only have $k$ non-zeros), we can compute a simple summation (which is a linear operator) of Shapley modules very efficiently. In fact, we never need to store the zero outputs of Shapley modules in practice and can merely maintain the output of $\tilde{f} : \mathbb{R}^{D \times C} \mapsto \mathbb{R}^{C''}$ via summation aggregation. For this sparse architecture, we can rewrite $g$ to emphasize the linear transformation we use:

$$\mathcal{G}_{\tilde{f}}(\boldsymbol{x}) = \text{sum}^{[D]} \circ \Omega \left( \boldsymbol{x}, \tilde{f} \right) = \text{sum}^{[D]} \circ g(\boldsymbol{x}; \tilde{f}) = \text{sum}^{[D]} \circ (A \Omega (\boldsymbol{x}, f))^T, \tag{5}$$

where $f$ is a vector-valued function, each $f_j$ have small active sets (i.e., $|\mathcal{A}(f_j)| \le k$), and the Shapley transform of $f_j$ is computed via Shapley modules. For any Shallow SHAPNET (including our sparse architecture based on Shapley modules and a linear transform), we have the following:

**Theorem 2.** *Shallow* SHAPNET*s compute the exact Shapley values, i.e.,* $g(\boldsymbol{x}; f) = \phi(\mathcal{G}_f, \boldsymbol{x})$.

We use Shapley modules inside both Shallow SHAPNETs and Deep SHAPNETs (subsection 2.3) to make our model scalable. This is similar in spirit with L-Shapley (Chen et al., 2019), which computes local Shapley values as an approximation to the true Shapley values based on the assumed graph structure of the data, but we build such locally computed Shapley values directly into the model itself.

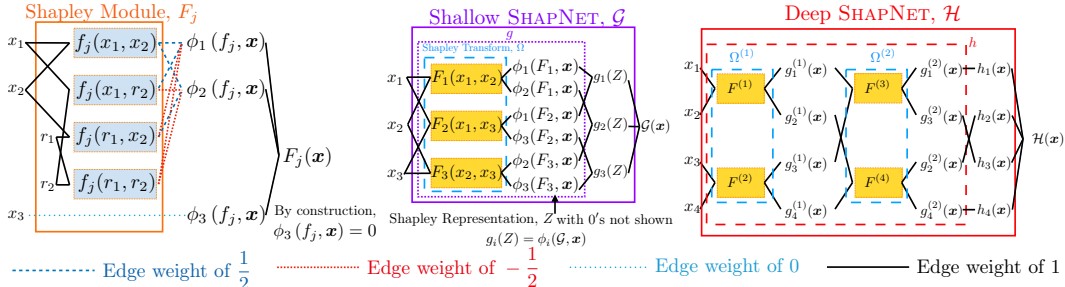

Figure 2: An example of how we construct SHAPNET from simple Shapley modules (left) which explicitly compute the SHAP explanation values for the arbitrary function $f_j$. Shallow SHAPNETs (middle) are based on computing many Shapley modules in parallel—in particular, we show computing all pairs of features—and then applying a summation to aggregate the modules (where implicit zeros are not computed). Deep SHAPNET (right) are composed of Shapley transform blocks where the output explanation of the previous layer is used as input for the next layer. We show an example using disjoint subsets for each Shapley module within a Shapley transform and then use a butterfly permutation (as in the Fast Fourier transform (FFT) algorithm) to enable complex dependencies as discussed in subsection 2.3. The edge weights shown here are direct results of Eqn. 1 with $d = 2$.

## 2.3 DEEP SHAPNET

While our Shallow SHAPNET is quite powerful, it does not enable inter-layer Shapley representations within a deep model. To enable this, we construct Deep SHAPNETs by cascading the output of Shapley transform from Def. 3, with the only restriction that the output of the last Shapley transform is the right dimensionality for the learning task (e.g., the output space of the last Shapley transform is $\mathbb{R}^{D \times c}$, where $c$ is the number of classes). Note that effectively the Shapley representation $Z$ in Def. 4 from a Shapley transform can be seen as input $\boldsymbol{x}$ to the next Shapley transform, making it possible to cascade the representations.

**Definition 8** (Deep SHAPNET). *A Deep SHAPNET $\mathcal{H}$ is based on a cascade of Shapley transforms:*

$$\mathcal{H} = \text{sum}^{[D]} \circ h = \text{sum}^{[D]} \circ \Omega^{(L)} \circ \Omega^{(L-1)} \cdots \circ \Omega^{(2)} \circ \Omega^{(1)}, \tag{6}$$

*where $h(\boldsymbol{x}) \in \mathbb{R}^{D \times c}$ is the Deep SHAPNET explanation, $c$ is the number of output classes or regression targets, $\Omega^{(l)}$ is the $l$-th Shapley transform with its own underlying function $f^{(l)}$, and all the reference values are set to 0 except for the first Shapley transform $\Omega$ (whose reference values will depend on the application).*

To ground our Deep SHAPNETs, we present the following theoretical properties about the explanations of Deep SHAPNETs (proof in Appendix B).

**Theorem 3.** *The Deep SHAPNET explanation $h(\boldsymbol{x})$ defined in Def. 8 provides an explanation that satisfies both local accuracy and missingness with respect to $\mathcal{H}$.*

**Deep SHAPNET with Disjoint Pairs and Butterfly Connections** It is immediately obvious that the computational graph determined by the choice of active sets in Shapley modules dictates how the features interact with each other in a Shallow SHAPNET and (since Shapley transform) a Deep SHAPNET. For our experiments, we focus on one particular construction of a Deep SHAPNET based on disjoint pairs (active sets of Shapley modules do not overlap: $\cap_{j=1}^{c} \mathcal{A}(f_j) = \emptyset$) in each layer and a butterfly permutation across different layers to allow interactions between many different pairs of features—similar to the Fast Fourier Transform (FFT) butterfly construction. This also means that the cardinalities of the active sets of all the Shapley modules are set to 2, making the overhead for computing the Shapley values roughly $4 \times$ that of the underlying function. An example of this construction can be seen on the right of Fig. 2. We emphasize that this permutation is one of the choices that enable fast feature interactions for constructing Deep SHAPNETs from Shapley transforms. We do not claim it is necessarily the best but believe it is a reasonable choice if no other assumptions are made. One could also construct SHAPNETs based on prior belief about the appropriate computational graph (an example for images below). Another possibility is to learn the pairs by creating redundancy and following by pruning as discussed in subsection 2.5 and Appendix E.

**Deep** SHAPNET **for Images**    Here we describe one of the many possibilities to work on image datasets with SHAPNETs, inspired by works including Desjardins et al. (2015); Dinh et al. (2017); Kingma & Dhariwal (2018). Still, the Deep SHAPNET here consists of different layers of Shapley transforms. We begin by describing the operations in each of the consecutive Shapley transforms, and then cascade from one stage to the next (see subsection I.3 for full details).

The canonical convolution operation is composed of three consecutive operations: sliding-window (unfolding the image representation tensor to acquire small patches matching the the filter), matrix multiplication (between filters and the small patches from the images), and folding the resulting representation into the usual image representation tensor. We merely 1) replace the matrix multiplication operation with Shapley modules (similar to that in Desjardins et al. (2015)) and 2) put the corresponding output vector representation in space $\mathbb{R}^{c'}$ back in the location of the original pixel (and performed summation of the values overlapping at a pixel location just as canonical convolution operation would), since we have the same number of pixels. To create a hierarchical representation of images, we use à-trous convolution (Yu & Koltun, 2016) with increasing dilation as the Shapley transform goes from one stage to the next, similar to Chen et al. (2017b); Chen et al. (2018). To reduce computation, we can choose a subset of the channel dimensions of each pixels to take as inputs, similar to Dinh et al. (2017); Kingma & Dhariwal (2018). For the final prediction, we do a global summation pooling so that the final representation conforms to the number of output needed.

## 2.4    EXPLANATION REGULARIZATION DURING TRAINING

One of the important benefits to SHAPNETs is that we can regularize the explanations *during training*. The main idea is to regularize the last layer explanation so that the model learns to attribute features in ways aligned with human priors–e.g., sparse or smooth. This is quite different from smoothing the explanation *post-hoc* as in saliency map smoothing methods (Smilkov et al., 2017; Sundararajan et al., 2017; Yeh et al., 2019), but falls more into the lines of using of interpretations of models as in Ross et al. (2017); Liu & Avci (2019); Erion et al. (2019); Rieger et al. (2020); Tseng et al. (2020); Noack et al. (2021). For $\ell_1$ regularization, the main idea is similar to sparse autoencoders in which we assume the latent representation is sparse. This is related to the sparsity regularization in LIME (Ribeiro et al., 2016) but is fundamentally different because it actually changes the learned model rather than just the explanation. Note that the explanation *method* stays the same. For $\ell_\infty$ regularization, the idea is to smooth the Shapley values so that none of the input features become too important individually. This could be useful in a security setting where the model should not be too sensitive to any one sensor because each sensor could be attacked. Finally, if given domain knowledge about appropriate attribution, different regularizations for different tasks can be specified.

## 2.5    CONDITIONAL DYNAMIC PRUNING

Corollary 4 allows for pruning during inference time and learning during training time (Appendix E).

**Corollary 4** (Missingness in Deep SHAPNETs). *If $Z_i^{(\ell)} = r_i^{(\ell)}$, then $Z_i^{(m)} = 0$ for all $m > \ell$, where $i$ is an index of a certain feature, and $Z_i^{(\ell)}$ and $r_i^{(\ell)}$ are the output and reference value respectively for the $\ell$-th Shapley transform.*

Corollary 4 (with proof in Appendix C) is a simple extension of Theorem 3: if a feature's importance becomes **0** at any stage, the importance of the same feature will be **0** for *all* the stages that follow. In particular, the computation of a Shapley module could be avoided if the features in its active set are all zeros. In other words, we can dynamically prune the Shapley modules if their *inputs* are all zeros. In practice, we set a threshold close to zero, and skip the computations involving the input features with importance values lower than the threshold. Note that the importance values for the features are conditioned on the input instances instead of a static pruned model (Ancona et al., 2020), allowing more flexibility. Moreover, with the help of $\ell_1$ regularization discussed in subsection 2.4 (which can applied on each $Z^{(l)}$), we can encourage more zeros and thus avoid even more computation. An illustration can be found in Fig. 3 where we mask the pixels with small $\ell_1$ norm after normalization, and pruning results are shown against overall performance of the model in Fig. 5. We draw similarity to prior works including Sun et al. (2013); Zeng et al. (2013); Han et al. (2015); Bacon (2015); Ioannou et al. (2016); Bengio et al. (2016); Theodorakopoulos et al. (2017); Bolukbasi et al. (2017).

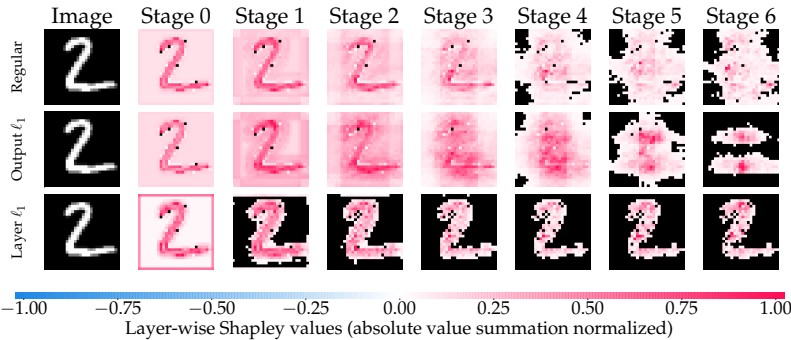

Figure 3: A visualization of Corollary 4 and pruning in action: Notice 1) that once a pixel get to close to reference values at an earlier stage, it will stay that way for the rest of the computational graph and 2) $\ell_1$ regularization promotes the sparsity and hence decreases the amount of computation required. The first row is retrieved from regularly trained model, the second row from model with $\ell_1$ regularization on the last layer discussed in subsection 2.4, and the last row from the model with $\ell_1$ regularization applied on *all* inter-layer representations.

## 3 EXPERIMENTS & VISUALIZATIONS

We will (1) validate that our SHAPNET models can be quite expressive despite the intrinsic explanation design, (2) demonstrate that our intrinsic SHAPNET explanations perform comparably or better than post-hoc explanations, and (3) highlight some novel capabilities. More details in Appendix I.

**Datasets**    First, we create a synthetic regression dataset by sampling the input $v \in [-0.5, 0.5]^{16}$ from the uniform distribution where the true (synthetic) regression model is $f(v) = \prod_i v_i + \sum_i v_i^2 + 0.05\epsilon$, where $\epsilon$ is sampled from a standard normal distribution. Second, we choose five real-world datasets from Dua & Graff (2017): Yeast ($d = 8$) and Breast Cancer Wisconsin (Diagnostic) ($d = 30$), MNIST (LeCun & Cortes, 2010), FashionMNIST (Xiao et al., 2017), and Cifar-10 (Krizhevsky, 2009) datasets to validate SHAPNET on higher-dimensional datasets.

SHAPNET **model performance**    We first validate that our Deep and Shallow SHAPNET models can be comparable in performance to other models—i.e., that our Shapley module structure does not significantly limit the representational power of our model. Thus, we define DNN models that are roughly comparable in terms of computation or the number of parameters—since our networks require roughly four times as much computation as a vanilla DNN with comparable amount of parameters. For our comparison DNNs, we set up a general feedforward network with residual connections (He et al., 2016). Note that for the vision tasks, we compare with two other models with our Deep SHAPNETs: a comparable convolutional-layer based model which shares the same amount of channels, layers, and hence parameters with only LeakyReLU activation (Xu et al., 2015) with no other tricks other than normalizing the input and random cropping after padding (same as our Deep SHAPNETs), and the state-of-the-art on each dataset (Byerly et al., 2021; Tanveer et al., 2020; Foret et al., 2021). The performance of the models are shown in Table 1 and Table 2 in which the loss is shown for the synthetic dataset (lower is better) and classification accuracy for the other datasets (higher is better). While there is a very slight degradation in performance, our lower-dimensional models are comparable in performance even with the structural restriction. While there is some gap between state-of-the-art for image data, our image-based Deep SHAPNET models can indeed do reasonably well even on these high-dimensional non-linear classification problems, but we emphasize that they provide fundamentally new capabilities as explored later. See Appendix I.

**Explanation quality**    We now compare the intrinsic SHAPNET explanations with other post-hoc explanation methods. First, because our intrinsic SHAPNET explanations are simultaneously produced with the prediction, the explanation time is merely the cost of a single forward pass (we provide a wall-clock time experiment in Appendix F). For the lower-dimensional datasets, we validate our intrinsic explanations compared to other SHAP-based post-hoc explanations by computing the difference to the true SHAP values (which can be computed exactly or approximated well in low dimensions). Our results show that Shallow SHAPNET indeed gives the true SHAP values up to

Table 1: Model performance (loss for synthetic or accuracy for others, averaged over 50 runs).

| Models / Datasets | Deep SHAPNET | DNN (eq. comp.) | DNN (eq. param.) | Shallow SHAPNET | GAM |
|---|---|---|---|---|---|
| Synthetic (loss) | 3.37e-3 | 3.93e-3 | 6.62e-3 | 3.11e-3 | 3.36e-3 |
| Yeast | 0.585 | 0.576 | 0.575 | 0.577 | 0.597 |
| Breast Cancer | 0.959 | 0.966 | 0.971 | 0.958 | 0.969 |

Table 2: Accuracies of Deep SHAPNETs for images, comparable CNNs and state-of-the-art models.

| Models / Datasets | Deep SHAPNET | Comparable CNN | SOTA |
|---|---|---|---|
| MNIST | 0.9950 | 0.9917 | 0.9984 |
| FashionMNIST | 0.9195 | 0.9168 | 0.9691 |
| Cifar-10 | 0.8206 | 0.7996 | 0.9970 |

Table 3: Average difference from exact Shapley values for different models.

| Explanations / Models | Ours | Deep SHAP | Kernel SHAP (77) | Kernel SHAP (def.) | Kernel SHAP (ext.) |
|---|---|---|---|---|---|
| Untrained models (100 independent trials each entry) | | | | | |
| Deep SHAPNET | 1.29e-06 | 2.89e-05 | 1.42e+03 | 2.13e+05 | 5.00e-11 |
| Shallow SHAPNET | 2.97e-07 | 0.802 | 0.0733 | 4.05e-03 | 1.05e-03 |
| Regression models on synthetic dataset (20 independent trials each entry) | | | | | |
| Deep SHAPNET | 4.99e-03 | 3.71 | 4.91e-03 | 3.05e-04 | 2.26e-05 |
| Shallow SHAPNET | 4.19e-08 | 3.88 | 4.05e-03 | 2.39e-04 | 2.75e-05 |
| Classification models on Yeast dataset (50 independent trials each entry) | | | | | |
| Deep SHAPNET | 0.0504 | 0.307 | 0.01467 | 0 | 0 |
| Shallow SHAPNET | 2.20e-07 | 0.516 | 0.0132 | 0 | 0 |
| Classification models on Breast Cancer Wisconsin (Diagnostic) (50 independent trials each entry) | | | | | |
| Deep SHAPNET | 0.0167 | 0.120 | 0.0369 | 4.09e-03 | 8.05e-04 |
| Shallow SHAPNET | 1.20e-04 | 0.0581 | 0.0224 | 1.91e-03 | 2.03e-04 |

Table 4: Explanation regularization experiments with Deep SHAPNETs (averaged over 50 runs).

| Models / Metrics | Yeast | | | Breast Cancer Wisconsin | | |
|---|---|---|---|---|---|---|
| | $\ell_\infty$ Reg. | $\ell_1$ Reg. | No Reg. | $\ell_\infty$ Reg. | $\ell_1$ Reg. | No Reg. |
| Coefficient of variation for abs. SHAP | **0.768** | 1.23 | 1.05 | **1.28** | NaN | 2.04 |
| Sparsity of SHAP values | 0.003 | **0.00425** | 0.00275 | 0.429 | **0.841** | 0.209 |
| Accuracy | **0.592** | **0.592** | 0.587 | 0.957 | **0.960** | **0.960** |

numerical precision as proved by Theorem 2, and Deep SHAPNET explanations are comparable to other post-hoc explanations (result in Table 3 and more discussion in Appendix G). We will evaluate our high-dimensional explanations in the MNIST dataset based on 1) digit flipping experiments introduced in Shrikumar et al. (2017), where higher relative log-odds score is better, 2) removing the least-$k$ salient features, where lower fraction of change in the output is better, 3) dropping the top-$k$ salient features and measuring the remaining output activation of the original predicted class, where ideally, the top features would drastically reduce the output logits—showing that our explanations do highlight the important features. We compare, with the same baselines for all values, Deep SHAPNET explanations with DeepLIFT (rescale rule in Shrikumar et al. (2017)), DeepSHAP (Lundberg & Lee, 2017), Integrated Gradients (Sundararajan et al., 2017), Input×Gradients (Shrikumar et al., 2016) and gradients (saliency as in Baehrens et al. (2010); Simonyan et al. (2014)). From Fig. 4, Deep SHAPNET offers high-quality explanations on different benchmarks.

**Layer-wise explanations** With the layer-wise explanation structure, we can probe into the Shapley representation at each stage of a Deep SHAPNET as in Fig. 3 and perform dynamic pruning. Hence we also provide a set of pruning experiment with retaining accuracy shown in Fig. 5, visualization shown in Fig. 3, and discussion in Appendix H, where we show how pruning the values in the Shapley representations of different layer changes model performance, and argue that a large portion of the computation can be avoided during inference time and the model can retain almost the same accuracy. This showcases the possibility of dynamic input-specific pruning. We note that though this has been done before, our models is, to the best of our knowledge, the only model capable of learning the importance of different computational components in tandem with the importance of each feature, performing conditional pruning during inference time and explanation regularization.

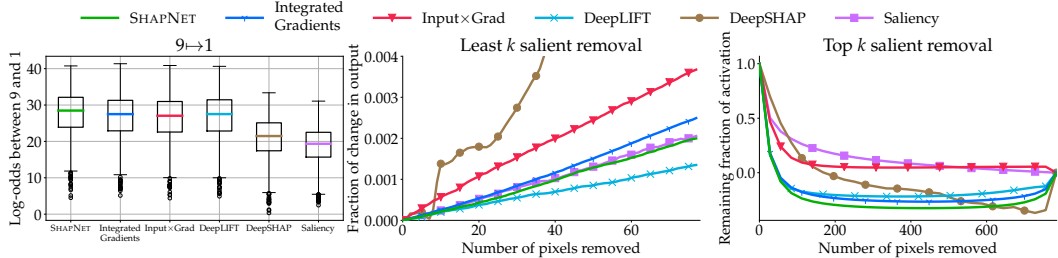

Figure 4: Our intrinsic Deep SHAPNET explanations perform better than post-hoc explanations in identifying the features that can flip the model prediction or that contribute most to the prediction as in figures on the left showing the results of flipping digits as introduced in Shrikumar et al. (2017) and right showing the remaining activation after removing the top $k$ features identified by each explanation method. While Deep SHAPNET explanations did not perform the best in the middle where we show the results after removing least $k$-salient features as introduced in Srinivas & Fleuret (2019), our model still scores the second. All results are measured on MNIST test set. More results for digit flipping, in Fig. 11, show the same conclusion with statistical significance in Table 7.

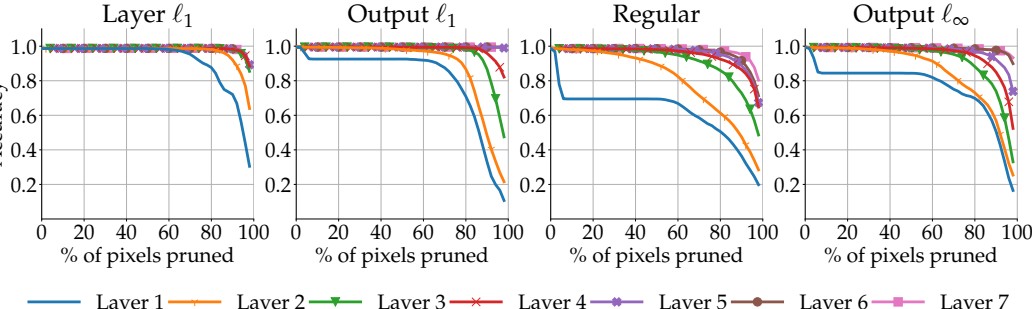

Figure 5: 1) Pruning values from earlier layers have at least equal or stronger effects on the performance than from later layers due to Corollary 4; 2) $\ell_1$ regularized model performs better especially when the regularization is applied on all all the layers instead of just the output; 3) Most of the values can be removed and retain the accuracy. We prune the values in the order from the least salient (in magnitude of $\ell_1$ norm) to the most pixel by pixel. This is measured on the MNIST test set.

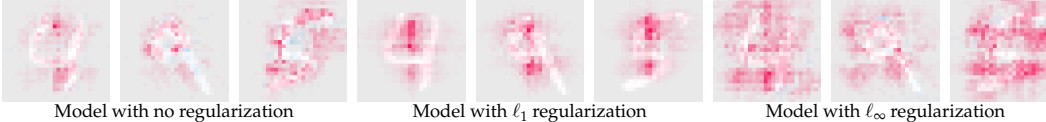

Figure 6: MNIST SHAPNET explanations for different regularizations qualitatively demonstrate the effects of regularization. We notice that $\ell_1$ only puts importance on a few key features of the digits while $\ell_\infty$ spreads out the contribution over more of the image. Red and blue correspond to positive and negative contribution respectively. Details in subsection I.4.

**Explanation regularization during training** We experiment with $\ell_1$ and $\ell_\infty$ explanation regularizations during training to see if the regularizations significantly affect the underlying model behavior. For the lower-dimensional datasets, we compute the accuracy, the sparsity, and the coefficient of variation (CV, standard deviation divided by mean) for our Deep SHAPNET. We see in Table 4 that our $\ell_\infty$ regularization spreads the feature importance more evenly over features (i.e., low CV), $\ell_1$ regularization increases the sparsity (i.e., higher sparsity), and both either improve or marginally degrade the accuracy. For MNIST data, we visualize the explanations from different regularization models in Fig. 6, and we perform the top-$k$ feature removal experiment with different regularizers in Fig. 7, which shows that our regularizers produce some model robustness towards missing features. Fig. 5 shows $\ell_1$ regularized model achieves the best result under pruning across different layers, especially when applied on all layers, further increasing the amount of computation to be pruned.

ACKNOWLEDGMENTS

R.W. and D.I. acknowledge support from Northrop Grumman Corporation through the Northrop Grumman Cybersecurity Research Consortium (NGCRC) and the Army Research Lab through contract number W911NF-2020-221. X.W. acknowledges support from NSF IIS #1955890.

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

## A PROOF OF THEOREM 2 & LEMMA 1

*Proof of Lemma 1.* For each row of $W \triangleq AZ$, we can show that it is the Shapley value for $\tilde{f}^{(k)}$ using the linearity of Shapley values (the original motivating axiom of Shapley values, see Shapley (1953, Axiom 4 & Axiom 5)):

$$\boldsymbol{w}_k = \sum_j a_{k,j} \phi(f^{(j)}, \boldsymbol{x}) = \phi(\sum_j a_{k,j} f^{(j)}, \boldsymbol{x}) = \phi(\tilde{f}^{(k)}, \boldsymbol{x}). \tag{7}$$

Combining this result for all rows with Def. 3, we arrive at the result. □

Now we are ready to present the proof for Theorem 2:

*Proof.* For a general $f$, we have that $G_f(\boldsymbol{x}) = f(\boldsymbol{x})$ because $\Omega(f, \boldsymbol{x})$ produces the Shapley values of $f$ by the definition of Shapley transform $\Omega$ (Def. 3) and Shapley values maintain the local accuracy property, i.e., $\sum_{i \in \mathcal{I}_D} \phi_i(f, \boldsymbol{x}) = f(\boldsymbol{x})$. Thus, $g(\boldsymbol{x}; f) = \Omega(\boldsymbol{x}, f) = \Omega(\boldsymbol{x}, G_f)$, which is equivalent to the Shapley values of $G_f$. (Our proposed sparse SHAPNET architecture, denoted by $\tilde{f}$, based on Shapley Modules and a simple linear transformation is merely a special case of the underlying function $f$.) □

## B PROOF OF THEOREM 3

*Proof.* The proof of local accuracy is trivial by construction since $\mathcal{H}(\boldsymbol{x}) \triangleq \text{sum}^{[D]}(h(\boldsymbol{x}))$ from Def. 8. The proof of missingness is straightforward via induction on the number of layers, where the claim is that if $x_i = r_i$ (i.e., the input for one feature equals the reference value), then $\forall \ell, \ h_i^{(\ell)}(\boldsymbol{x}) = 0$ (i.e., the pre-summation outputs of the Shapley modules in all layers corresponding to feature $i$ will all be zero). For the base case of the first layer, we know that if $x_i = r_i$, then we know that $Z_i^{(1)} = \boldsymbol{0}$ by the construction of Shapley modules. For the future layers, the inductive hypothesis is then that if previous layer representation $Z_i^{(\ell-1)}$ of a feature $i$ is $\boldsymbol{0}$, then the representation from the current layer $\ell$ for this feature remains the same, i.e., $Z_i^{(\ell)} = \boldsymbol{0}$. Because the reference values for all layers except the first layer are zero, then we can again apply the property of our Shapley modules and thus the inductive hypothesis will hold for $\ell > 1$. Thus, the the claim holds for all layers $\ell$, and the end representation $h(\boldsymbol{x})$ will also be zero—which proves missingness for Deep SHAPNETs. □

## C PROOF FOR COROLLARY 4

*Proof.* Given the existence of a feature index $i \in \mathcal{I}_C$ and a layer index $\ell$ where $Z_i^{(\ell)} = \boldsymbol{0}$, where note that $\boldsymbol{0}$ is the reference values we set for inter-layer representations (see subsection 2.3), we can truncate the model to create a sub-model that starts at layer $\ell$, which, by the definition of missingness, will ensure, by the same proof as in Appendix B, the corresponding place in the representation to be zero (i.e., $\forall m > \ell, Z_i^{(m)} = \boldsymbol{0}$). □

## D DEDICATED LITERATURE REVIEW

### D.1 EXPLANATION METHODS

Our methodology sits *in between* post-hoc and intrinsic approaches by constructing an explainable model but using SHAP post-hoc explanation concepts and allowing more complex feature interactions.

**Post-hoc feature attribution** Post-hoc feature-attribution explanation methods gained traction with the introduction of LIME explanations (Ribeiro et al., 2016) and its variants (Ribeiro et al., 2018; Guidotti et al., 2018a;b), which formulate an explanation as a local linear approximation to the model near the target instance. LIME attempts to highlight which features were important for the prediction. Saliency map explanations for image predictions based on gradients also assign importance to each feature (or pixel) (Simonyan et al., 2014; Shrikumar et al., 2017). SHAP explanations attempt

to unify many previous feature attribution methods under a common additive feature attribution framework (Lundberg & Lee, 2017; Lundberg et al., 2020), which is the main motivation and foundation for our work.

Layer-wise Relevance Propagation (LPR) (Bach et al., 2015) provides a way to decompose the prediction into layered-explanations, similar to our layer-wise explanation with the caveat that our inter-layer explanation corresponds to concrete input features.

From a neighbouring route, gradient-based saliency map is another family of methods to highlight important components of each input (Ancona et al., 2018). Input gradients (saliency as in Baehrens et al. (2010); Simonyan et al. (2014)) tries to identify the important features for a prediction value by taking the gradient from the output w.r.t the input. Input×Gradients (Shrikumar et al., 2016) uses the saliency map acquired by the input gradient method and performed element-wise multiplication with the input instance, which tried to sharpen the saliency map. Integrated Gradients (Sundararajan et al., 2017), on the other hand, sets up a reference values from which the gradient is integrated to the current input values, and integrate the input gradient taken along a chosen path from the reference values to the current input instance.

**Intrinsically interpretable models** Intrinsically interpretable models are an alternative to post-hoc explanation methods with a compromise on expressiveness of the models. In particular, GAM (Lou et al., 2012) and its extension (Lou et al., 2013; Caruana et al., 2015; Chen et al., 2017a; Wang et al., 2018; Agarwal et al., 2020) construct models that can be displayed as series of line graphs or heatmaps. Because the entire model can be displayed and showed to users, it is possible for a user to directly edit the model based on their domain knowledge. In addition, Alvarez Melis & Jaakkola (2018) proposed an architecture that are complex and explicit if trained with a regularizer tailored to that architecture.

## D.2 APPROXIMATING SHAPLEY VALUES

To alleviate the computational issue, several methods have been proposed to approximate Shapley values via sampling (Strumbelj & Kononenko, 2010) and weighted regression (Lundberg & Lee, 2017, Kernel SHAP), a modified backpropagation step (Lundberg & Lee, 2017, Deep SHAP), utilization of the expectation of summations (Ancona et al., 2019, DASP), or making assumptions on underlying data structures (Chen et al., 2019, L-Shapley and C-Shapley). In terms of computational complexity, Sampling based methods (Strumbelj & Kononenko, 2010; Lundberg & Lee, 2017) are asymptotically exact in exponential complexity in the number of input features. Deep Approximate Shapley Propagation (DASP) approximates in polynomial time with the help of distributions from weighted summation of independent random variables (von Bahr, 1972) and assumed density filtering (Boyen & Koller, 1998; Gast & Roth, 2018), L-Shapley and C-Shapley do in linear with the Markov assumption on the underlying data structure, Deep SHAP approximates with an additional backward pass after the forward pass, and our method performs approximation with a constant complexity with a single forward pass, which additionally allows regularizing the explanations according to human priors and hence modifying the underlying model during training.

## D.3 REGULARIZING ON THE EXPLANATIONS

Our SHAPNETs allow the users to modify the underlying function based on the explanations. This falls in line with previous works (Ross et al., 2017; Liu & Avci, 2019; Erion et al., 2019; Rieger et al., 2020; Tseng et al., 2020), each of which adds an extra term in the loss function measuring the disagreement between the explanations and human priors. Ross et al. (2017) proposed to regularize the input gradients as explanations, which creates the need for second-order gradient. Liu & Avci (2019) uses Integrated Gradients (Sundararajan et al., 2017, IG), which is an extension to Shapley values (Sundararajan & Najmi, 2020). Due to the use of IG, several samples (for one explanation) are additionally warranted. Erion et al. (2019) proposed Expected Gradient to rid of reference values in IG and suggested the usage of a single sample. Tseng et al. (2020) proposed to force the model to focus on low-frequency component of the input data by applying Fourier Transform on input gradients. Rieger et al. (2020) adds to this line of work with Contextual Decomposition (Murdoch et al., 2018; Singh et al., 2019) which avoids the need of computation of second-order gradients and extra samples, and in addition enable regularization on interactions. Our work differs with

these five in that we do not need a second order gradient and that we are focusing on the more theoretically-grounded Shapley values.

The explanation-related regularizations in Alvarez Melis & Jaakkola (2018); Plumb et al. (2020); Noack et al. (2021) differ with ours in that the regularizers are used to *promote* interpretability and ours uses regularization to modify the underlying model to conform to prior belief. (Noack et al. (2021) aims at connecting interpretability with adversarial robustness. Continuing that line of work, Plumb et al. (2020) proposed EXPO, a regularizer to help improve the interpretability of black boxes. EXPO was designed not for a specific structure but for all neural network models. We also draw close similarity to works that that attempt to smooth saliency map explanations (Smilkov et al., 2017; Sundararajan et al., 2017; Yeh et al., 2019) which acts on explanation methods to induce certain properties in explanations but not the actual models.

### D.4 CONDITIONAL PRUNING

In contrast to prior work where people perform static structure pruning based on Shapley values (Ancona et al., 2020), we compute the importance values *on the fly* during inference time.

**Conditional Computation in Deep Neural Networks**   Conditional computation refers to a certain fashion where the neural network only activates a certain subset of all computations depending on the input data (Bengio et al., 2016). This is desirable in low-power settings where the device can run only a fraction of the entire model to save memory. Works investigating this computation style include Sun et al. (2013); Zeng et al. (2013); Han et al. (2015); Bacon (2015); Ioannou et al. (2016); Bengio et al. (2016); Theodorakopoulos et al. (2017); Bolukbasi et al. (2017).

## E   LEARNED FEATURE INTERACTIONS

With both $\ell_1$ regularization and missingness in Deep SHAPNET as discussed in subsection 2.5, we can discuss how it is possible to learn the specific choices for Shapley modules in each Shallow SHAPNET in a Deep SHAPNET (how features interact in a Deep SHAPNET). Note that we have already discussed how we can perform instance-based pruning during inference.

To learn the interactions, we first create redundancies in each Shapley transform inside a Deep SHAPNET. During training, we automatically gain the importance values for each features by construction, from which we will determine the modules that are worth computing by comparing input representation of the features in their active sets to $0$s. If the input representation is close enough to $0$s, we can simply ignore that module. Of course, the underlying function can always change during training, and we argue that we can slowly strip away modules from the latter layers to the first layers as the training progresses to allow for more expressiveness and more room to correct early mistakes by the training procedure.

## F   WALL-CLOCK TIME FOR EXPLANATION EXPERIMENT

To validate the efficiency of computing an explanation, we compare the wall-clock time of our SHAPNET explanations to other approximation methods related to Shapley-values including Deep SHAP and Kernel SHAP with different numbers of samples—40, 77, and default ($2 \times d + 2^{11}$). The wall-clock computation time in seconds can be seen in Table 5. We note importantly that this comparison is a bit odd because other Shapley-value explanation methods are *post-hoc* explanations, whereas SHAPNET is both a model and an intrinsic explanation method. Thus, the most fair comparison in terms of time is to have the post-hoc explanation methods explain our SHAPNET model as seen in the first column of Table 5. Nevertheless, we also show the wall-clock times for the post-hoc methods on our comparable DNNs in columns two and three of Table 5—note that SHAPNET cannot explain other DNNs hence the N/A's. From the first column, we can see that our method is inherently faster than the others when applied to our model. However, smaller model (i.e., DNN w. equivalent parameters) may come with faster explanations.

Table 5: Time for computing explanations (averaged over 1000 runs)

| Explanations \ Models | Deep SHAPNET | DNN (eq. comp.) | DNN (eq. param.) |
|---|---|---|---|
| Ours | 20.47 | N/A | N/A |
| Deep SHAP | 46.10 | 83.38 | 8.56 |
| Kernel SHAP (40) | 60.78 | 130.04 | 11.52 |
| Kernel SHAP (77) | 72.61 | 201.88 | 13.53 |
| Kernel SHAP (def.) | 598.79 | 789.08 | 142.85 |

## G   LOWER-DIMENSIONAL COMPARISON BETWEEN SHAPNET EXPLANATIONS AND POST-HOC EXPLANATIONS

For lower-dimensional datasets, we seek validation for the explanations $h(\boldsymbol{x})$ provided by Deep SHAPNETs by comparing to the true Shapley values of our models (which can be computed exactly or aproximated well in low dimensions). Our metric is the normalized $\ell_1$ norm between the true Shapley values denoted $\phi$ and the explanation denoted $\gamma$ defined as:

$$\ell(\phi, \gamma) = \frac{\|\phi - \gamma\|_1}{d \sum_i \phi_i}, \tag{8}$$

where the number of dimension $d$ is used as an effort to make the distance Note that we measure the average difference from vector-output classifiers by means discussed in subsection G.1. The experimental setup and results are presented in subsection G.2.

### G.1   COMPARING SHAP APPROXIMATION ERRORS WITH VECTOR OUTPUT MODELS

To compare the approximation errors for every output of the model, we use a weighted summation of the explanations for different outputs, where the weights are the models' output after passing into the soft-max function. Thus, we up weight the explanations that have a higher probability rather than taking a simple average. Thus, in the extremes, if the predicted probability for a label is zero, then the difference in explanation is ignored, while if the predicted probability for a label is one, then we put all the weight on that single explanation.

Concretely, for a classification model $C : \mathbb{R}^d \mapsto \mathbb{R}^n$, where $d$ is the number of input features and $n$ is the number of classes, we have the output vector of such model with an input instance $\boldsymbol{x} \in \mathbb{R}^d$ as $C(\boldsymbol{x}) \in \mathbb{R}^n$. For each output scalar value $[C(\boldsymbol{x})]_j$, we compute the approximation of the SHAP values w.r.t. that particular scalar for feature indexed $i$, denoted by $\gamma_i^{(j)}$, and hence its corresponding errors as measured in normalized $\ell_1$ distance as discussed in Eqn. 8: $\ell_1^{(j)}$. The final error measure that we compare between classifiers $\ell_1^*$ is simply a weighted sum version of the normalized $\ell_1$:

$$\ell_1^* = \sum_{j=1}^n [\text{softmax}(C(\boldsymbol{x}))]_j \cdot \ell_1^{(j)},$$

where the softmax is taken on the non-normalized output of the classifiers. Note that $\ell_1^j$ is also taken on the raw output.

### G.2   SETUP AND RESULTS

Because the explanation difference from SHAP is independent of the training procedure but solely depends on the methods, we compare the performance of the explanation methods on untrained models that have randomly initialized parameters (i.e., they have the model structure but the parameters are random)—this is a valid model it is just not a model trained on a dataset. Additionally, we consider the metric after training the models on the synthetic, yeast, and breast cancer datasets. Note that for even a modest number of dimensions, computing the exact Shapley values is infeasible so we use a huge number of samples ($2^{16}$) with the kernel SHAP method to approximate the true Shapley values as the ground truth for all but the Yeast dataset, which only requires $2^8$ samples for exact computation. Note that the extended Shapley value (ext.) requires $2^{14}$ times of model evaluation for a single explanation which is already extremely expensive, and thus included merely for comparison.

Table 6: Standard deviation of Shapley approximation errors on different datasets

| Explanations / Models | Ours | DeepSHAP | Kernel SHAP (77) | Kernel SHAP (def.) | Kernel SHAP (ext.) |
|---|---|---|---|---|---|
| Untrained models | | | | | |
| Deep SHAPNET | 0.734 | 5.23 | 7.67e+08 | 4.25e+09 | 9.20e-03 |
| Shallow SHAPNET | 7.55e-07 | 1.57 | 0.124 | 8.02e-03 | 2.98e-03 |
| Regression models on synthetic dataset | | | | | |
| Deep SHAPNET | 3.28e-03 | 5.15 | 3.13e-03 | 8.79e-05 | 2.23e-05 |
| Shallow SHAPNET | 3.09e-04 | 1.28 | 4.75e-03 | 1.62e-04 | 6.33e-06 |
| Classification models on Yeast dataset | | | | | |
| Deep SHAPNET | 0.105 | 0.358 | 0.0288 | 0 | 0 |
| Shallow SHAPNET | 5.35e-07 | 0.894 | 0.0304 | 0 | 0 |
| Classification models on Breast Cancer Wisconsin (Diagnoistic) | | | | | |
| Deep SHAPNET | 0.0104 | 0.0359 | 0.0987 | 3.61e-03 | 1.72e-03 |
| Shallow SHAPNET | 0.0109 | 0.246 | 0.0141 | 1.93e-03 | 2.91e-04 |

Results of the average difference can be seen in Table 3. From the results in Table 3, we can see that indeed our Shallow SHAPNET model produces explanations that are the exact Shapley values (error by floating point computation) as Theorem 2 states. Additionally, we can see that even though our Deep SHAPNET model does not compute exact Shapley values, our explanations are actually similar to the true Shapley values.

We also provide in Table 6 the standard deviation of the approximation errors as in Table 3. These are the same runs as those in Table 3.

## H    PRUNING EXPERIMENT

We also conduct a set of pruning (set to reference values) experiment. This is, again, enabled by the layer-wise explainable property of Deep SHAPNET. Recent work has investigated the usage of Shapley values for pruning (Ancona et al., 2020; Ghorbani & Zou, 2020), but our work means to showcase the ability even further. Results are shown in Fig. 5.

We perform the pruning experiment with all four MNIST-traiend models as discussed in subsection 2.3 and subsection 2.5 with no, output $\ell_1$, layer $\ell_1$, and $\ell_\infty$ regularizer, respectively. The values are removed (set to reference) by the order of their magnitude (from smallest to largest). We can see that most of the values might be dropped and the model performance stays unhinged. Moreover, $\ell_1$ seems to have the best robustness against pruning, which makes sense as it induces sparsity in explanation values. Future works are to be done on the computational cost reduction under such pruning technique.

## I    MORE EXPERIMENTAL DETAILS

For all the experiments described below, the optimizers are Adam (Kingma & Ba, 2014) with default parameters in PyTorch (Paszke et al., 2019).

### I.1    TIMING EXPERIMENTS

For Table 5, the timing is for the untrained (randomly initialized) models with 16 features. The numbers after Kernel SHAP indicates the number of samples allowed for Kernel SHAP's sampling method. Time is measured in seconds on CPU with 1000 independent trials per cell as Kernel SHAP supports solely CPU from the implementation of the authors of Lundberg & Lee (2017). The model of the CPU used for testing is Intel(R) Xeon(R) Silver 4114 CPU @ 2.20GHz.

We perform explanation on randomly initialized 16-feature Deep SHAPNET model for 1000 rounds. The 16-feature setup means we will have $(16/2) \times \log_2 16 = 32$ Shapley modules inside the entire

model spanned out in 4 stages according to the Butterfly mechanism. The inner function $f$ for every module in this setting is set as a fully-connected neural network of 1 hidden layer with 100 hidden units. For the two reference models, the model equivalent in computation time has 8 layers of 1800 hidden units in each layer, while the model equivalent in parameters has 11 layers of 123 hidden units in each layer. The output of the models are all scalars.

## I.2 YEAST AND BREAST CANCER DATASETS EXPERIMENT SETUPS

This setup applies for the experiments involving both datasets, including those experiments for model performance and those that showcase the ability for explanation regularization. The experiments on the two datasets were run for 10 rounds each with different random initialization and the end results were obtained by averaging. For each round, we perform 5-fold cross validation on the training set, and arrive at 5 different models, giving us 50 different models to explain for each datasets. For the two datasets, we do 9 training epochs with a batch size of 32.

For the preprocessing procedures, we first randomly split the data into training and testing sets, with 75% and 25% of the data respectively. Then we normalize the training set by setting the mean of each feature to 0 and the standard deviation to 1. We use the training sets' mean and standard deviation to normalize the testing set, as is standard practice.

The inner functions of the Shapley modules are fully connected multilayer neural networks. The first and second stages of two models on the two datasets are identical in structure (not in parameter values). The first stage modules have 2 inputs and 25 outputs, with a single hidden layer of size 50. The modules in the second stage have 100 input units and 50 output units with two hidden layer of size 100. The non-linearity is introduced by ReLU (Nair & Hinton, 2010) in the inner functions. The rest of the specifics about the inner function are discussed below. Note that the dimensionality of the output from the last stage differs with that of the input to the next stage, as is expected by Shapley module where the input has twice the dimensionality of that of the output.

**Model for Yeast dataset**  The Yeast dataset has 8 input features and hence 4 Shapley modules at each of the $\log_2 8 = 3$ stages. Within each stage, the structure of the inner function is identical except for the parameters. For the Yeast dataset, the third (last) stage comprises four-layer fully-connected model with inputs from $50 \times 2 = 100$ units to the number of classes, which is 10, with two hidden layers of 150 units.

**Model for Breast Cancer Wisconsin (Diagnostic)**  This dataset has 30 input features, which is not a exponent of 2, but we can still construct a Deep SHAPNET by setting two extra variables to zero. In fact, in doing so we can simplify the Shapley modules since we know which of the modules in the network always have one or both of its input being 0 at all times. To construct the model, we realize that the number of stages is 5 and the number of modules at each stage is 16. For the Breast Cancer Wisconsin (Diagnostic) dataset, in the the third stage of the model, we use a model with 100 inputs units and 75 output units with two hidden layers of 150 units. The fourth stage has input units of $75 \times 2 = 150$ and output of 100 units with two hidden layers of size 200. The fifth (last) stage has a input dimension of 200, two hidden layers of 250 units and output of size 10.

## I.3 SPECIFICS FOR VISION TASKS

### I.3.1 STRUCTURE USED FOR VISION EXPERIMENTS

For all of the vision tasks, we use the same structure with the exception of input channel counts. We use a simple Linear-LeakyReLU (Xu et al., 2015) inside each Shapley Module, and the following are the output dimension of the linear layer: 1, 128, 128, 128, 128, 128, 128, 10, with 1 being the input channel of images in the case of MNIST & FashionMNIST or 3 with Cifar-10.

**À-trous convolution**  À-trous convolution is used for learning hierarchical representations of images. In our model, the dilations are set as follows: 1, 3, 5, 7, 9, 11, 13.

Table 7: The boost in performance is statistically significant as, shown below, the t-scores for SHAPNETs vs. different methods

| Explanations / Digits | Degrees of Freedom | Integrated Gradients | Input×Gradients | DeepLIFT | DeepSHAP | Saliency |
|---|---|---|---|---|---|---|
| $9 \mapsto 1$ | 1008 | 28.0404 | 40.2129 | 28.7084 | 118.6641 | 101.5194 |
| $4 \mapsto 1$ | 981 | 54.4659 | 68.6895 | 63.1221 | 139.4350 | 122.9292 |
| $8 \mapsto 3$ | 973 | 56.8543 | 67.6736 | 56.8097 | 128.9730 | 94.6011 |
| $8 \mapsto 6$ | 973 | 41.5245 | 57.2258 | 49.9137 | 97.5942 | 90.1728 |

### I.3.2 TRAINING RECIPE

**Learning process** As discussed before, all the models are trained with Adam in PyTorch with the default parameters. The batch size is set to 64 for all the vision tasks. No warm-up or any other scheduling techniques are applied. The FashionMNIST & MNIST models in Table 1 are trained for 10 epochs while the MNIST models with which we investigate the new capabilities (layer-wise explanations & Shapley explanation regularizations) in Section 3 are trained with just 5 epochs. The Cifar-10 models are trained for 60 epochs.

**Preprocessing** For both MNIST dataset and FashionMNIST dataset, we normalize the training and testing data with mean of 0.1307 and standard deviation of 0.3081, this gives us non-zero background and we would therefore set the reference values to 0's, allowing negative or positive contribution from the background. Both FashionMNIST and MNIST share the same preprocessing pipeline: 1) for training dataset, we extend the resolution from $28 \times 28$ to $36 \times 36$ and then randomly crop to $32 \times 32$, and then normalize. 2) for testing data, we simply pad the image to $32 \times 32$ and normalize them.

For Cifar-10 dataset, during training, we first pad 4 pixels on four sides of the image and then randomly crop back to $32 \times 32$, perform a horizontal flip with probability 0.5, and normalize the data with means $0.4914, 0.4822, 0.4465$ and standard deviaiton of $0.2023, 0.1994, 0.2010$ for each channel, respectively. For testing, we simply normalize the data with the same mean and standard deviation.

### I.4 ILLUSTRATIONS OF EXPLANATION REGULARIZATION EXPERIMENTS

We present visual representations of our models' explanation on MNIST (LeCun & Cortes, 2010) dataset in Fig. 8, Fig. 9, and Fig. 10. In all three figures, the gray images from left to right are explanations laid out for all 10 classes from 0 to 9, where the red pixels indicate positive contribution and blue pixels indicate negative contribution, with magnitudes presented in the color bar below.

From an intuitive visual interpretation, we argue that our explanation regularizations are indeed working in the sense that $\ell_\infty$ regularization is indeed smoothing the explanations (to put more weight on a large number of pixels) and that $\ell_1$ regularization limits the number of pixels contributing to the final classification.

One can notice that, by comparing the color bars below the figures, the contribution magnitudes for $\ell_\infty$ and $\ell_1$ regularized models are lower than those of the unregularized counterpart. We have expected this to happen as both regularization techniques require to make one or more of the attribution values to be smaller. However, we note that this change in scale of the attribution values should not impact the accuracy of the prediction as the softmax operation at the end will normalize across the outputs for all of the classes.

More interesting discussions are provided in the caption of Fig. 8, Fig. 9, and Fig. 10, and we strongly encourage readers to have a read.

### I.5 MORE RESULTS ON DIGIT FLIPPING

More results on digit flipping are in Fig. 11. We also provide the t-scores and the degrees of freedom (instead of p-values because it is so statistically significant that p-values are extremely close to 0's.) of the paired t-tests to show that the explanation methods of ours provide boost in performance that is statistically significant in Table 7.

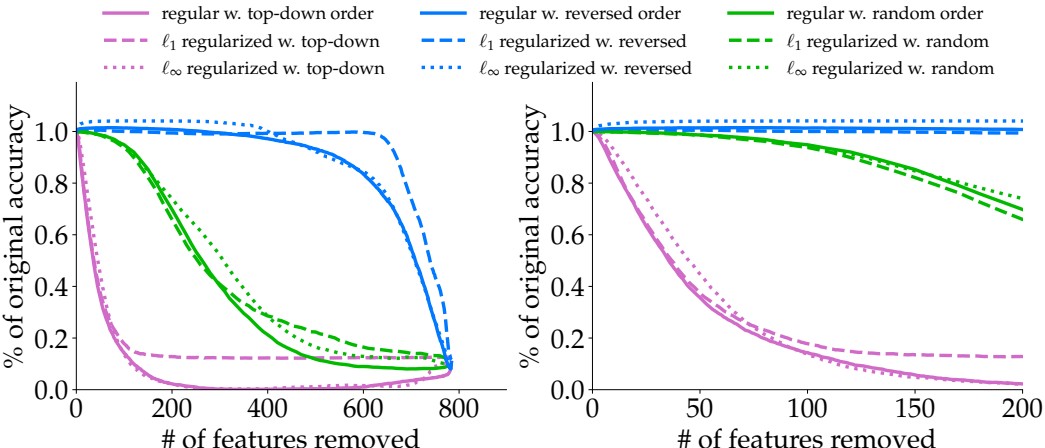

Figure 7: Both $\ell_1$ and $\ell_\infty$ regularizations often introduce some robustness to the model against feature removal (the dotted and dashed lines are often above the solid lines). Three different feature orders are used: 1) most positive to most negative (top-down), 2) most negative to most positive (reversed), and 3) randomly chosen as a baseline. $\ell_1$ regularization puts most importance on the first 100 features with relatively low importance to other features (seen in both top-down and reversed). $\ell_\infty$ regularization improves robustness when removing the top 50 or so features (as seen in the right figure which is zoomed in on the top 200 features).

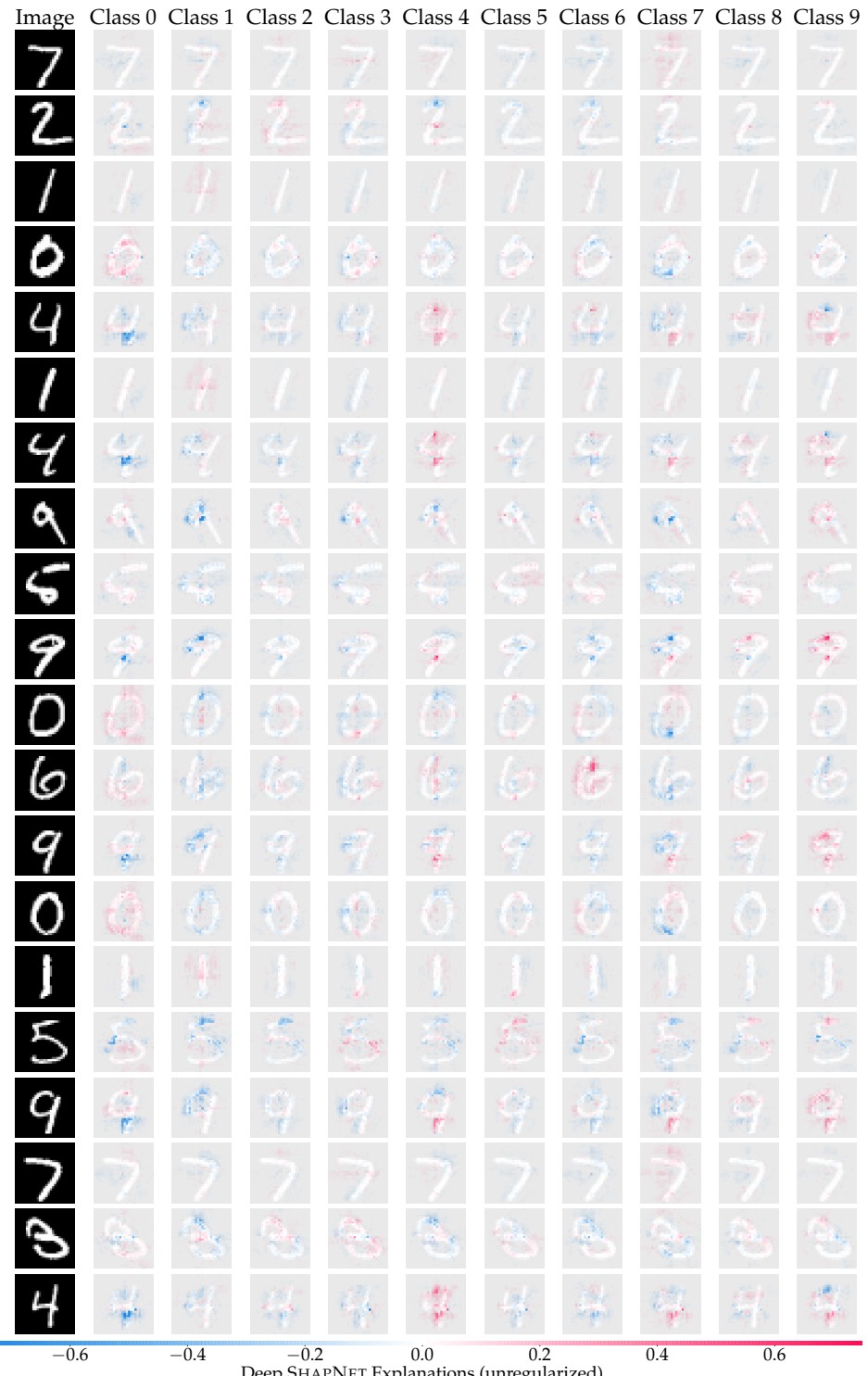

Figure 8: The explanations produced by our model trained without explanation regularization for all 10 classes. From left to right are class index (also corresponding digits) from 0 to 9.

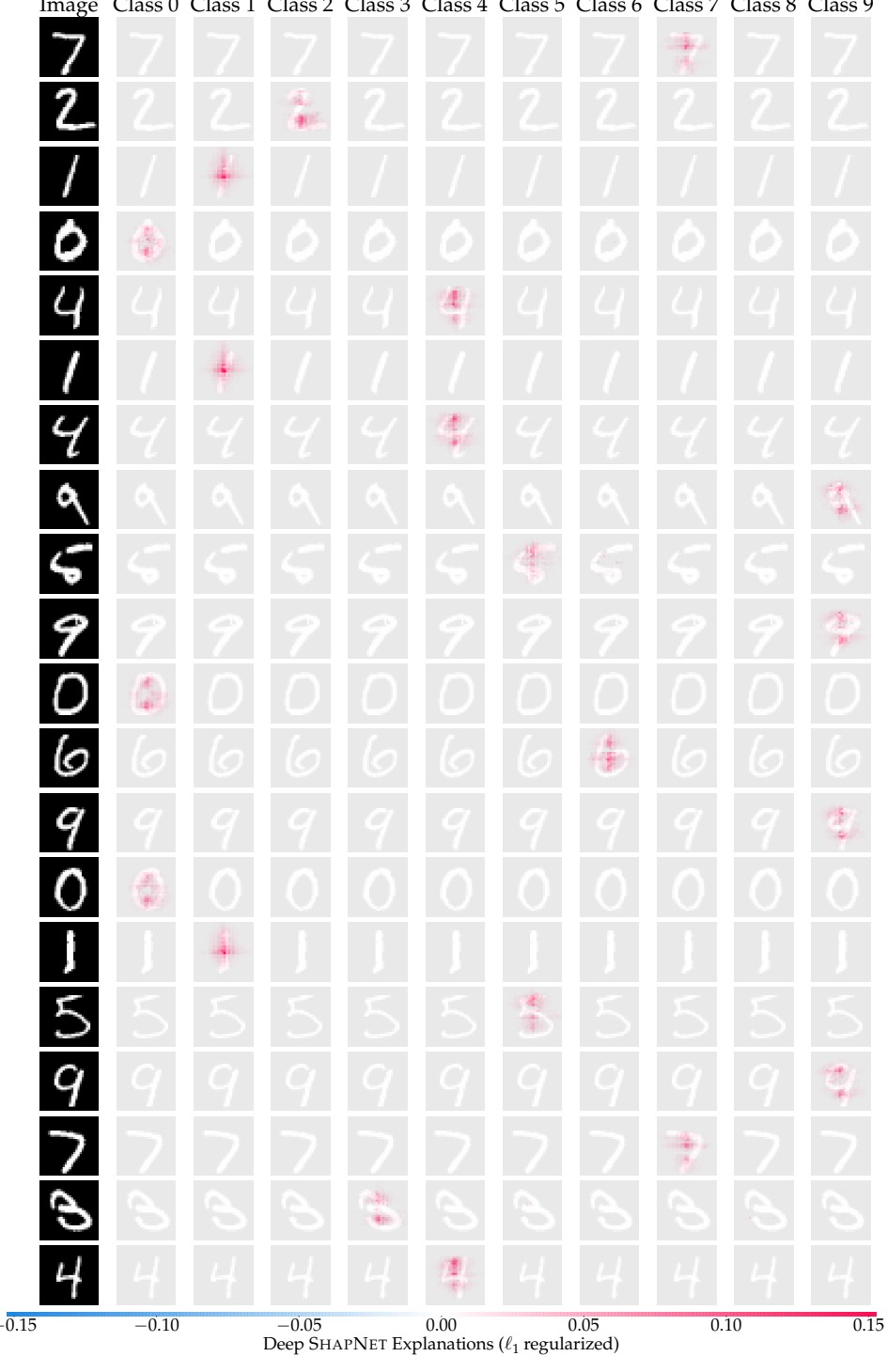

Figure 9: $\ell_1$ regularization promotes sparsity in feature importance, which, by comparison with Fig. 8 and Fig. 10, one can easily argue has been successfully achieved. This shows that $\ell_1$ regularization, combined with cross entropy loss, does (almost) remove negative contribution and only focuses on each digit's distinguishing features.

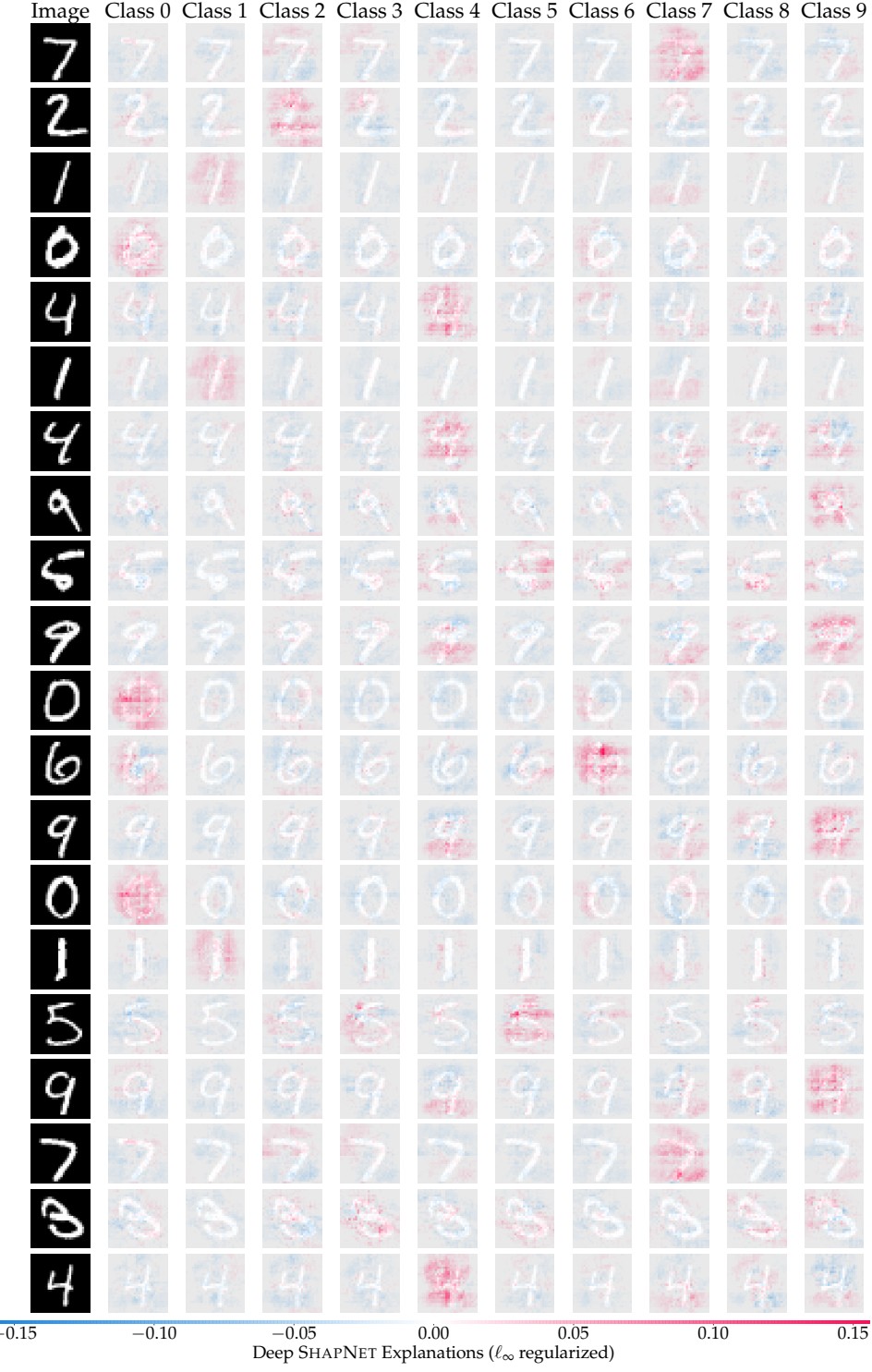

Figure 10: With $\ell_\infty$ regularization, the contribution among pixels are more laid out and less concentrated than they would be other wise as shown in Fig. 8. The explanations produced by our model trained with $\ell_\infty$ explanation regularization for all 10 classes. However, it is still easy to notice which part of the images contribute positively or negatively for a particular class.

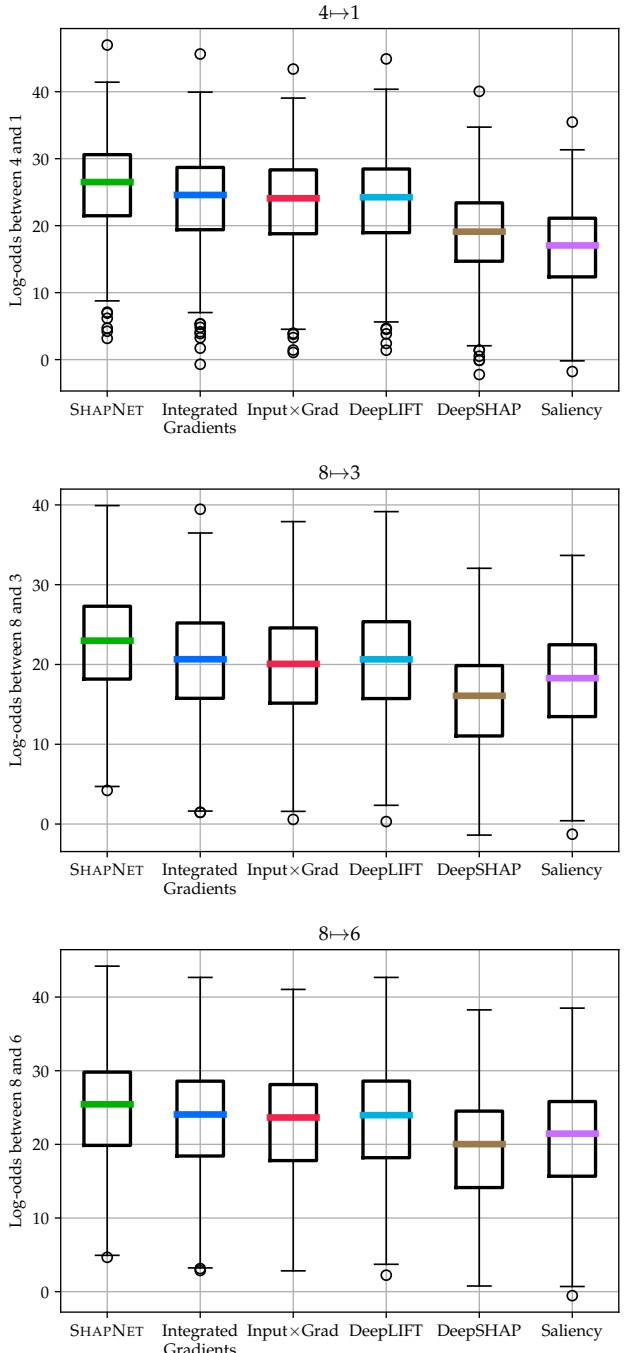

Figure 11: Digit flipping results: Deep SHAPNET explanations perform well against post-hoc methods.

