# OpenReview forum: "Shapley Explanation Networks"
_ICLR.cc/2021/Conference — ICLR 2021 Poster_

### Official Review · AnonReviewer3 · 2020-10-27
**Nice idea, but limiting the cardinality of active sets of the Shapley modules seems to be restrictive**

**Rating:** 6
**Confidence:** 3

**Review:**

##########################################################################

Summary:

The paper proposes to incorporate Shapley values as latent representations in deep models. Specifically, the paper constructs Shallow SHAPNETs that computes the exact Shapley values. The paper also constructs Deep SHAPNETs that maintain the missingness and accuracy properties of Shapley values. The effectiveness of the proposed SHAPNETs is demonstrated through experiments on synthetic and real-world data.

Overall, it seems to be a good idea to incorporate Shapley values into deep models and the proposed method seems to be reasonable. The empirical results have demonstrated the usefulness of the proposed method. The paper is also well-written and technically sound. I have some comments as detailed below.

##########################################################################

Comments:

- The main challenge for Shapley values is its computational complexity. The paper overcomes this challenge by forcing the active set of all Shapley modules to be size=2. While mitigating the computational challenge, this would limit the representation power of the model. The authors showed that this is not a big issue by providing a comparison in Table 1 on three datasets (Synthetic, Yeast, Breast Cancer). However, all three datasets are low dimensional and do not require a high representation power of the model. Therefore, I am not quite convinced that the proposed SHAPNETs have satisfactory representation power. A comparison of Deep SHAPNETs and the DNN models on the three high-dimensional image datasets (MNIST, FashionMNIST, Cifar-10) would better answer this question.

- In principle, we can tradeoff between the representation power and the computational efficiency by varying the size of the active sets of the Shapley modules. Have the authors considered a comparison of SHAPNETs with different active set sizes?

- In Table 1, Shallow SHAPNET has a better performance than the Deep SHAPNET. Why is that?

---

> ### Author Response · Authors · 2020-11-17
> **We thank Reviewer 3 for the positive feedback and comments**
>
> 1. To alleviate concerns about representational power, we have updated our original model performance results for image datasets to include comparisons to comparable CNN classifiers and state-of-the-art classifiers.  We have placed these new results in Table 2 (note that the original manuscript contained results for image datasets on the right of Table 1but did not include comparisons).  As the results show, the difference between our Deep ShapNets and a comparable CNN is minimal.  Although there is some drop in performance compared to highly tuned state-of-the-art classifiers, the Deep ShapNet for images retains reasonable performance even for these complex high-dimensional datasets.  While we admit this is not a perfect performance, we suggest these results show that ShapNet can indeed be quite expressive, and fine-tuning may be able to close this gap.  We also suggest two reasons why our Deep ShapNets could be expected to have comparable performance.  First, our channels and meta-channels (unified as channel dimensions in the new manuscript) for each feature that needs to be explained enable quite rich intermediate representations similar to standard convolutional networks.  Second, we apply a shared non-linear function across the spatial dimensions, which is similar to a convolution that applies a shared linear function across spatial dimensions.  (If we have misunderstood your comment, please correct our understanding.  We admit that we were slightly confused by the comment regarding image datasets because we included image classifier performance in the original manuscript–albeit maybe easy to miss–though we hope the additional comparisons will make this clear.)
> 2.  We thank the reviewer for suggesting a comparison of different active size sets.  We will consider an experiment to explicitly show this but may not have time to implement so we mention a few things that could alleviate the concern. First, we note that the construction of Deep ShapNet for images uses active sets with a cardinality of 4 (2×2 kernel size in the convolution neural network language) instead of 2, which did allow higher power (Cifar-10 had 40% accuracy with a cardinality of 2 but was able to get 85% with a cardinality of 4).  Second, we note that it is somewhat challenging to isolate the effects of different active set sizes because the underlying functions must also be different (e.g., the input size is 2 vs 4 so do you increase the hidden dimension of the underlying functions or keep the same).  Overall, we think a deeper exploration of this trade-off could be interesting future work.
> 3.  Thanks for the question.   The  Shallow  ShapNet construction has overlapping  Shapley modules,  e.g.,  for a  dataset with feature dimensionality of  16,  we would have $16×(16−1)/2=  120$  pairwise  Shapley  Modules,  which is the number of all possible pairs.  However,  our Deep ShapNets used a Butterfly-mechanism-like structure,  which as a result has$16/2×log_2{16} = 32$ Shapley Modules in total.  In this sense, if the number of features needed for interaction is low in a dataset, it might be possible and reasonable for Shallow ShapNet to outperform Deep ShapNet based on the number of Shapley Modules they have respectively.

---

### Official Review · AnonReviewer4 · 2020-10-28
**Clever idea, but empirical results could be stronger**

**Rating:** 7
**Confidence:** 4

**Review:**

(I am raising my score by 2 points after the author response)

--

SUMMARY

The authors present a new type of network architecture where the each intermediate layer outputs the shapely contributions of the input $\boldsymbol{x} \in \mathbb{R}^d$ (with respect to some reference $\boldsymbol{r} \in \mathbb{R}^d$)  to $\sum_{j=1}^c f^{(j)}(\boldsymbol{x})$, where $f^{(j)}$ are functions that are parameterized by a neural network.  The functions $f^{(j)}$ can have vector-valued outputs (called "meta-channels" and denoted by the letter $n$ in the text), in which case the layer would have output dimensions $\mathbb{R}^{n \times d}$ (one entry for each meta-channel and input feature). Typically, computing shapely contributions to a function requires a number of operations that increases exponentially with the input size; the core trick that the authors use to make their networks scalable is to aggressively limit the subset of features (which they call the "active set") that are provided to each individual function $f^{(j)}$ - specifically, if only two inputs $(x_1, x_2)$, with reference values $(r_1, r_2)$, are provided to $f^{(0)}$, then the shapely contributions $\phi(f^{(0)}, x_1)$ and  $\phi(f^{(0)}, x_2)$ can be computed exactly from $f^{(0)}(x_1, x_2), f^{(0)}(x_1, r_2), f^{(0)}(r_1, x_2)$ and $f^{(0)}(r_1, r_2)$. In the case of deeper networks, the "active sets" (i.e. inputs) of the functions $f^{(j)}$ are disjoint pairs of size 2 that cover the full set of inputs, and from one layer to the next the pairs are permuted, which can allow the network to learn complex dependencies. When computing shapely contributions to subsequent layers, the "meta-channels" for a given input feature are grouped together - thus, every layer outputs estimated Shapely contributions for the original $d$ input features. To obtain the network output, the shapely contributions after the final network layer are simply summed over the $d$ input features. The authors also extend this setup to work on computer vision tasks, where parameters are shared between different input patches.

The authors benchmark their networks on two real-world tabular datasets and several computer vision tasks, and claim that their proposed Shapely Explanation Networks, or ShapNets, can achieve competitive performance while retaining explainability. On the tabular datasets, the authors demonstrate that their Shapely Explanation Networks provide good estimates of the true Shapely contributions for a fraction of the computational cost compared to methods like KernelSHAP. The authors also demonstrate how the explanations provided by ShapNets can be used for attribution regularization.

STRENGHTS

- I think the idea of these ShapNets is very interesting, and the authors have clearly put considerable thought into how to design the ShapNets to make them efficient and scalable. The authors do well to draw a comparison to generalized additive models, which are easy to interpret but do not learn interactions; these ShapNets could potentially enable good explainability while also allowing the network to learn interactions
- Estimating Shapely values through ShapNets is far more computationally efficient that KernelSHAP (table 3 of the supplement) and more accurate (with respect to the true Shapely values) than DeepSHAP, as shown in Table 4 of the supplement (however, agreement with the true Shapely values is not synonymous with the explanation being the most useful in a particular practical situation, as I discuss below)
- Because the Shapely values are output by the network, they can also be regularized during model training

WEAKNESSES

(1)  The authors mention that they use an FFT-like butterfly permutation in order to allow the network to learn complex interactions between pairs of inputs, and they set the number of layers to be log_2 [number of features]. As diagrammed on the right of Figure 2, it does not look like this permutation is sufficient to allow every input to interact meaningfully with every other input - in particular, $x_1$ does not seem to be interacting (strongly) with $x_4$, and $x_2$ does not seem to be interacting (strongly) with $x_3$. The reason I put "strongly" in parentheses is because the only way $x_1$ can interact with $x_4$ is by modifying the Shapely contribution $g_2^{(1)}(x)$ of the input $x_2$ - but in the case where $x_2$ does not vary much (e.g. imagine it is always set to its reference value), that would altogether prevent the network from learning an interaction between $x_1$ and $x_4$. Also, it is counter-intuitive for a network to learn an interaction between two features by modifying the Shapely contribution of a third feature. It seems like the core issue is the absence of a hierarchical structure? The use of log_2 [number of features] layers implies a binary-tree-like hierarchical structure that seems to be missing in this setup.

(2) One thing that can limit the applicability of ShapNet is that, for tabular data, TreeSHAP already provides a way to learn a powerful predictive model while also providing an efficient way to compute the SHAP values. Thus, in order to make a case for ShapNet on tabular inputs, the authors would need to show that ShapNet outperforms tree-based models. Table 1 shows the results of benchmarking on the Yeast and Breast Cancer datasets, but it looks like Generalized Additive Models achieve the best accuracy on these datasets (outperforming the DNNs as well as ShapNet)...thus, it seems that these datasets don't have complex interactions in them, and thus there wouldn't be a reason to prefer ShapNets over GAMs on these datasets (let alone prefer ShapNets over tree-based models).

(3) Of course, tree-based models are not suitable for computer vision data, and ShapNets definitely have an edge-here - however, the benchmarking of explanation quality for the computer vision data seems to be limited to Figure 3, which used the "drop in accuracy as features are removed" as measure of explanation quality. This particular metric has issues in that the modified image could be out-of-distribution (see the ROAR paper by Hooker et al. - https://arxiv.org/abs/1806.10758). I would recommend that the authors instead either explore: (a)  the "remove least k salient pixels" method from the FullGrad paper: https://arxiv.org/pdf/1905.00780.pdf (b)  the "digit flipping" experiments used in the DeepSHAP/DeepLIFT papers, or (c) (best) train their models on the BAM dataset and quantify interpretation quality accordingly: https://github.com/google-research-datasets/bam. Also, I would be interested in seeing the timing benchmarks for these computer vision models (it seems that Table 3 in the supplement is for an untrained randomly initialized model with 16 features). I would also be curious to see how the contributions estimated by ShapNets compare to the true Shapely values for the computer vision datasets (Table 4 in the supplement seems to report this for tabular data only). However, I understand if it is too computationally expensive to accurately estimate the true Shapely values on vision data.

(4) A note regarding Table 4 in the supplement: while it is good that ShapNets produce explanations that are similar to the true Shapely values (as this is their stated goal), we should not mistake "agreement with true Shapely values" as meaning the explanation is useful in a particular practical application - for that, different benchmarks would be required (such as the ones mentioned in the previous point).

(5) The authors present "Layer-wise explanations" as a "new capability", but this seems a bit misleading; any post-hoc explanation method can be applied to give explanations for intermediate layers by simply treating the intermediate layer as the first layer of a truncated model, and using the activations of the intermediate layer under the reference input as the reference value of the intermediate layer (e.g. see https://captum.ai/api/neuron.html).

(6) In a similar vein, the authors present "Explanation regularization during training" as a "New capability" but this is a bit misleading; explanation regularization during training has been performed using post-hoc attribution methods (as the authors acknowledge under "Extended Literature Review" in Appendix C). The authors don't seem to have compared their proposed explanation regularization during training to these earlier works.

(7) The paper was very hard to read - it was only thanks to staring at Figure 2 that I managed to form a sense of what was going on, and I think most readers would similarly struggle. One point of confusion is that there are two kinds of outputs of each layer - the function output, denoted by the capital letter $G$ in Definition 11, as well as the shapely contributions, denoted by the lowercase $g_i$. The authors don't always clearly distinguish which "output" is being considered (e.g. the authors write "we construct deep ShapNets by cascading the output of Shallow ShapNets from Def. 11", but Def. 11 is for the function output denoted by the capital letter $G(x)$, whereas from Figure 2 it is clear that what is provided to subsequent layers is not $G(x)$, but rather the Shapely contributions $g_i(x)$.  Similarly, the discussion of "meta channels" is very confusing; when"meta channels" are first introduced in Remark 9, the authors describe meta-channels as the result of applying a Shapely Transform to a tuple of vector-valued functions - i.e. it is implied that meta-channels arise as part of the outputs of the ShapNet layers; however, the example of meta-channels that is given is the RGB channel of pixels, which is in the input. I eventually gathered that meta-channels are analogous to what one would normally think of as the "channels/neurons" in a neural network, because the actual "channel dimension" as it is described in this paper disappears due to the summation over the channels $c$ in Definition 11. I think the flow of the paper would be greatly improved if the authors stated their big-picture strategy up-front: that they can compute Shapely values efficiently because every parameterized function only operates on two features, and that they sum over many such functions in order to obtain the layer output (the analogy to a generalized additive model was helpful).

MINOR

- The caption of Figure 3 should specify which model/task the lines are being plotted for
- The authors do not appear to have trained their models using early stopping; based on the methods description in the supplement (section G), it seems that the models were trained for fixed numbers of epochs. This does not, as far as I can tell, change the core results of the paper, but it is important context (e.g. helpful to know when evaluating  the change in accuracy due to regularization in Table 2).
- In table 4, the authors wrote 1.42e+03 for KernelShap & Deep SHAPNet for the untrained models - I'm guess that was meant to be 1.42e-03?
- The authors write "DeepLIFT (scaled to get DeepSHAP)" - can they clarify what they mean by this scaling? Are they referring to the Rescale rule of DeepLIFT? The Rescale rule doesn't refer to a scaling of the explanations; it's a particular type of modified backpropagation rule that was described in the DeepLIFT paper.
- A suggestion regarding explanations that are computed with respect to the softmax logits, in case the authors are not doing this already: for each feature, it may be helpful to subtract the average contribution score across all classes from the contribution score to each individual class; that way, you can account for the softmax normalization, in that if a feature contributes equally to the logits of all classes then it effectively contributes to none of the classes (due to cancellation in the softmax). This point is discussed in the DeepLIFT paper in the section "Adjustments for Softmax Layers".

EXPLANATION OF RATING

While I think the core idea of ShapNets is very interesting and promising, I think the empirical results are currently somewhat lacking - in particular, I hope to see a clear use-case where one would prefer ShapNets over alternative methods. For this reason, I'm rating the paper at marginally below the acceptance threshold.

---

> ### Author Response · Authors · 2020-11-17
> **We thank the reviewer for bring up this issue and give several use-cases**
>
> Due to the space limitations, we first address the issues on the use case.
> 1. The feature that uniquely distinguishes ShapNets from other models or other explanation methods is that ShapNets have layer-wise explanations that assign, at each stage, importance to input features of the entire model (not the input to an intermediate layer of a model), which cannot be achieved if either the model or the explanation method is switched. Perhaps among other things, this unique property enables a fundamentally new capability: instance-based network pruning: given an input instance, as the computation progresses and at any layer, we can set the $k$ least-salient values in the representation to zeros. This operation could exclude the need to compute the Shapley value, by the missingness property, for an input importance value set to zero---hence the pruning.  In particular, the computation of a Shapley Module could be avoided if its features in active sets are all set to zeros. Further, in the same fashion that we proved the missingness property for Deep ShapNets, once the importance values for a given feature is set to zero at an earlier stage, all the importance values of that particular feature will no longer need computation.
> In a low power setting, the system can consecutively load each Shallow ShapNet layer into memory one by one.
> With our instance-based pruning, this system can choose a subset of Shapley Modules to load instead of all of them at once. Note that the choice of Shapley Modules varies from instance to instance, avoiding the need to load those responsible for other instances. Moreover, if trained with $\ell_1$ regularization, as shown in Fig. 7 in the original submission and Fig. 5 of the current submission, more of the values and hence their later-stage computation can be pruned. We have provided our pruning experiments in Fig. 7 in the original manuscript and have included it as Fig. 5 in the updated manuscript
> 2. Another of the key use-cases for our approach is the ability to regularize the model using explanations. While we were only able to explore a few simple regularizations, we think a model that intrinsically produces explanations will enable more complex regularizations in the future. For example, a domain expert could provide some "ground-truth" attribution maps, and we could regularize the explanations toward these "ground-truth" attribution maps for the inputs that have these annotations. If annotations are not available, we could use superpixels of each image to create image-dependent group-wise explanation regularization for each image (i.e., all attribution values in a superpixel should be similar).
> 3. Another use-case could be when explanations are required on resource-constrained devices such as medical imaging in a remote setting or military intelligence in the field. Most post-hoc explanation methods (particularly for images), would require either a large amount of computation (e.g., KernelShap) or large memory (e.g., DeepShap that requires a backward pass through the network and thus intermediate activations must be saved in memory). Our approach would enable both prediction and explanation in a single forward pass without storing any intermediate activations. Our model (and thus explanation as well) could additionally be compressed just like any other prediction model for use on resource-constrained devices, not to mention the dynamic pruning described in the first point

---

> > ### Author Response · Authors · 2020-11-17
> > **The other issues**
> >
> > We will address the concerns the reviewer has in the order the reviewer raised them.
> > 1. We thank the reviewer for such detailed and thorough thoughts on our architecture design. While part of the concern on this permutation is indeed valid (as we will discuss next), the FFT butterfly enables at least weak interactions between all features in a very few numbers of layers. A simple way to increase the power of our Deep ShapNet is to apply the butterfly permutation multiple times and shuffle the coordinates in between each butterfly mechanism. We also note that the FFT permutation structure can be seen as multiple hierarchical trees put together (e.g., $x_1$ and $x_2$ combined to get $g^{(1)}_1$ and then $g^{(1)}_1$ and $g^{(1)}_3$ combined to get $g^{(2)}_1$). While further exploration of the architecture and permutations would definitely interesting, we did not find it necessary in our experiments and acknowledge that our FFT permutation introduces some bias in the model. Regarding the specific example by the reviewer, the example is indeed an interesting one though we do not believe this significantly limits the interaction power. Indeed, if $x_2$ is set to the reference value exactly (i.e., no variation), then $g^{(1)}_2(x)$ will be a constant, and therefore the model will not be able to learn an interaction. However, if there is even some (very) small variation of $x_2$, then the representation could produce a signal based on $x_1$ in $g^{(1)}_2$ that can be propagated to interact with $g^{(1)}_4$.  We do not fully understand why it is counterintuitive for the network to pass information through the interlayer Shapley values. On the one hand, we see that Shapley values are supposed to focus on the contribution of a single feature; however, given that we want to allow for complex interactions, this kind of information exchange is necessary and probably unavoidable unless you restrict to GAM models or similar.
> > 2.  On the GAM v.s. Shallow/Deep ShapNets: theoretically speaking, GAM can be viewed as a special case of Shallow ShapNet (by setting the active sets to each individual feature and only one feature) whose solution space encapsulates that of the GAM model. Therefore it is only natural that in the face of a complex dataset. The better performance from GAM is generally because of the chosen dataset. We will find more complex datasets for this, and perform a comparison with Tree-based models and TreeSHAP.
> > 3.  We thank the reviewer for suggesting further experiments, and we have updated the manuscript with the (a) remove least $k$ salient experiment (our method scored second) and (b) digit flipping experiment (our model scored first) in Fig. 4, and if the time of discussion phase allows, we will update with BAM results, and timing benchmarks for these CV models as well.
> > 4. We agree that closeness to Shapley values does not guarantee great performance in a practical sense, but given that Shapley values are already used in practice, we suggest that it is a nice property nonetheless.
> > 5.  We thank the reviewer for pointing the possibility of using post-hoc explanations for layerwise explanations. We do clarify two key differences from using a post-hoc approach. First, our layer-wise explanations are grouped to correspond with the original input features to the entire model, which means we have a concrete object to point to when assigning such importance. (For the post-hoc approach raised by the reviewer, the interpretation of the intermediate representations do not necessarily correspond to the original input features.) This, combined with missingness in Deep ShapNet, allows us to do dynamic pruning during inference time, where if explanations based on one pixel came sufficiently close to zeros, they will always remain zeros in the computation that follows. Note that this pruning is instance-based (or batch-based, if that is more efficient), which keeps the model and the model selects different computation graphs each time, which is different from canonical pruning techniques which end up with a different model. Second, we can compute all layerwise explanations with a single forward pass, which is clearly much more efficient than using post-hoc methods.  It is our mistake not highlighting such distinctions upfront, for which we apologize and have updated the manuscript. We have also removed the label "New capability" to avoid misleading readers.
> > 6.  Given the chance of misleading readers, we have removed the title "New capability" from the manuscript similar to above. However, we emphasize that our approach enables a natural intrinsic model explanation regularization. Importantly, our approach does not require calling a posthoc explanation method (which can be expensive) as a subroutine in the training algorithm, and our method based on Shapley values has arguably more theoretical grounding. Thus, our approach provides a unique and novel approach to intrinsic explanation regularization.

---

> > > ### Author Response · Authors · 2020-11-17
> > > **On the presentation of the idea**
> > >
> > > We thank the reviewer for spending time and effort on our work and raising proper concerns.
> > > We have revamped the notations and the order in which Shapley Modules, Shallow/Deep ShapNets are presented. In general, we first introduce Shallow ShapNet first, which computes Shapley values for each scalar of the vector-valued function output of the underlying function $f$, followed by introducing Shapley Module, which limits the number of features allowed to interact with each other at one time, as a way to limit the computational overhead, and lastly stack the Shallow ShapNets to form the Deep ShapNet. We have also taken care of the `situation as well as the ``meta-channel", which is not needed in the new notation any more. Hopefully this revision can make the work more understandable.
> > >
> > > Minor:
> > > 1.  The task in Figure 3 is on MNIST, which has been updated in the manuscript. Apologies for the confusion.
> > > 2. We thank the reviewer for pointing this out. In our setting, we wanted to keep as much of the setting fixed as possible while changing the regularization. Fixing the number of epochs is one way to isolate the effects of regularization as model performance is not the primary concern in demonstrating the effect of regularization.
> > > 3. As large as the value is, it is not a typo. This is because of one of the regularizers used in the implementation from the SHAP paper [1] (which works well when applied to trained models) seems to produce significant problems with untrained models.
> > > 4. We apologize for causing this confusion. In our current version, we compare our explanations with both DeepSHAP and DeepLIFT.
> > > 5. We thank the reviewer for this suggestion and has accordingly updated our figures in the appendix that contain visualizations of pixel importance of 20 MNIST test images for 10 classes for all three models (two regularized and one vanilla).
> > >
> > > We would like to express our gratefulness for the reviewer to read through our manuscript and our response.
> > >
> > > [1] Scott  M  Lundberg  and  Su-In  Lee.A  Unified  Approach  to  Interpreting  Model  Predic-tions.In I. Guyon, U. V. Luxburg, S. Bengio, H. Wallach, R. Fergus, S. Vish-wanathan,  and  R.  Garnett  (eds.),Advances  in  Neural  Information  Processing  Systems  30,pp. 4765–4774. Curran Associates, Inc., 2017.

---

> > > > ### Comment · AnonReviewer4 · 2020-11-25
> > > > **Thank you for addressing concerns**
> > > >
> > > > I thank the authors for their revised manuscript and am happy to raise my score by two points to be a 7 (this is still a preliminary score as I have not discussed the paper with other reviewers). I particularly appreciate that the authors rewrote the text to be more accessible, and I agree with the nuance that the authors' layer-wise explanations correspond to the original input features. I recognize there is not much time left in the discussion window, so consider the following to be feedback for the camera-ready version if the paper is accepted:
> > > >
> > > > - For the digit-flipping experiments, could the authors add p-values to verify whether their method is statistically significantly better?
> > > > - I see that the authors have included both DeepLIFT and DeepSHAP. Because there are many different implementations of DeepLIFT (and DeepSHAP used with a single baseline is identical to the 'Rescale' variant of DeepLIFT for many models), it would be helpful if they authors could clarify the specifics of which implementation of DeepLIFT was used, as well as the choice of baseline(s) for DeepLIFT and DeepSHAP.
> > > > - A minor note regarding the authors' comment "we do not fully understand why it is counterintuitive for the network to pass information through the interlayer Shapley values. On the one hand, we see that Shapley values are supposed to focus on the contribution of a single feature; however, given that we want to allow for complex interactions, this kind of information exchange is necessary and probably unavoidable unless you restrict to GAM models or similar" - I'm not quite sure why the authors write that this is "probably unavoidable" as it seems that by simply adding more layers of the butterfly permutation (as the authors suggest) in order to have a direct connection between the two features, the model would not need to use the inter-layer shapely values of a third feature to model these types of interactions (unless the ground-truth interaction is actually being mediated by the value of the third feature)

---

### Official Review · AnonReviewer2 · 2020-10-29
**Interesting new models for generating fast SHAP explanations**

**Rating:** 6
**Confidence:** 4

**Review:**

This work proposes a new model class designed to make SHAP value calculations more efficient. The proposed method exploits sparsity and additivity among intermediate values to provide fast exact SHAP values for shallow ShapNets, and fast approximate SHAP values for Deep ShapNets. This approach enables SHAP-based regularization during training, layer-wise explanations, and faster SHAP-based explanations with minimal loss in quality.

The approach seems promising, but I have several questions and concerns about the method and presentation. I'll start with questions about the method.

- The authors write that SHAP values provide the unique explanation that satisfies a number of desirable properties; however, as pointed out by many works at this point [1, 2, 3], there are several proposed approaches for handling the held out features. Why do the authors choose to use single reference values rather than the interventional or conditional distribution? To what extent is the ShapNet approach amenable to other definitions of missingness?
- A couple questions about the shallow ShapNet. (i) When performing a forward pass, is it actually necessary to calculate the SHAP values (the Shapley representation)? The sum of the SHAP values $\phi_i(f^{(j)}, x)$ for $i = 1, \ldots, d$ is equal to $f^{(j)}(x)$, so it seems that SHAP value computation (represented by $g$ or $\Omega$) is not required unless one requires the explanation (e.g., for regularization). If that's the case, then a follow-up question is (ii) could any function whose output is the linear combination of intermediate values $f^{(j)}(x)$ be called a shallow ShapNet? And does your version of a shallow ShapNet yield faster explanations only because of the sparsity in the intermediate value functions?
- Is the ShapNet method unable to work on models that are not the linear combination of intermediate values (e.g., the classification probability after softmax activation rather than logit)? If so, that would be worth mentioning briefly.
- It seems that the recommendation is to determine the sparsity pattern in the value functions $f^{(j)}$ using the butterfly trick inspired by FFT; are there any other approaches worth considering that could improve either performance or compute time? Do the authors think that it would be worth learning the pairs that achieve the best performance?
- Perhaps the most important question about Deep ShapNet explanations is how similar they are to SHAP explanations (e.g., from KernelSHAP). If possible, the metrics from Table 4 would ideally be shown in the main text, and perhaps extended to include results from the vision models. These metrics, along with Table 3, seem very promising.

Some concerns regarding the presentation and experiments:

- Strumbelj and Kononenko [4] proposed a sampling algorithm for SHAP values that should be cited in the second paragraph of the introduction, along with the other estimation algorithms.
- Figure 2 is helpful but a bit complicated due to all the lines and bounding boxes, and the legend does not exactly correspond to the colors used. This figure may be worth revising and clarifying.
- There's a lot of verbosity used to describe some straightforward ideas involving SHAP values. The "Shapley transform," the "Shapley representation" and the "Shapley module" are all based on the simple idea of concatenating SHAP values into a matrix/tensor. The authors might consider presenting these ideas in a more straightforward manner.
- The sum operator $\text{sum}^{[\alpha_i]}$ uses ambiguous notation. Shouldn't the superscript indicate the dimension to sum across rather than the number of entries along that dimension? It would be unclear how to apply the operator $\text{sum}^{[\alpha_i]}$ if there was ever $\alpha_i = \alpha_j$ for some $j$.
- The generalized Shapley transform $\mathcal{S}$ is not clearly defined in Definition 13. Perhaps the authors could provide a definition in terms of $\Omega$? The use of this notation in the ShapNet composition (Eq. 6) seems to mask an underlying dependence on functions like $f^{(1)}, \ldots, f^{(c)}$ at each layer. Would it be possible to use notation that shows these functions?
- In Table 1, perhaps baseline model performance should be provided for the image datasets as a reference.
- The text and caption for Figure 3 do not seem to indicate which dataset is used. The results look promising; could KernelSHAP be provided as an additional comparison method?
- The authors point out the ShapNets enable (i) explanation regularizations and (ii) layer-wise explanations. Table 2, Figure 4 and Figure 6 show that the regularization works as intended, but this is a lot of space to prove this point; other results might be more important to include in the main text (mentioned earlier). Figure 5 shows that layer-wise explanations are possible, but the text does not explain how this is useful or when users would want this.

Despite some concerns, I think this paper contributes some novel ideas that move us towards tractable + principled explanations for expressive models.

[1] Aas et al., "Explaining individual predictions when features are dependent: More accurate approximations to Shapley values" (2019)

[2] Kumar et al., "Problems with Shapley-value-based explanations as feature importance measures" (2020)

[3] Merrick and Taly, "The Explanation Game: Explaining Machine Learning Models Using Shapley Values" (2019)

[4] Strumbelj and Kononenko, "An Efficient Explanation of Individual Classifications using Game Theory" (2010)

##########
Update
##########

I'm pleased with many of the changes the authors have made in response to the reviewers' concerns. However, I'll mention a couple lingering concerns.

- The authors have updated their paper regarding use-cases for layer-wise explanations, but it remains unclear whether this feature is impactful. While it is possible to perform instance-level pruning with ShapNets, is there any reason to do so? Finding features with low SHAP values requires actually calculating the SHAP values, which takes longer than just evaluating the function. Therefore, it seems that this technique would not lead to saving either time or memory. If the authors disagree, they may consider improving this aspect of the paper.
- The authors have clarified that they use the "reference values" approach to holding features primarily because it is convenient. To my knowledge, no existing research actually advocates for this approach. Most research advocates for either the "interventional" [2] or "observational conditional" approach [1]. The authors cite a paper that mentions the "reference values" idea (Baseline Shapley) [3], but not even this work truly advocates for this approach. In my view, this is a rather severe limitation of the proposed approach, and the authors did not suggest that they see a way to overcome it. To call these "SHAP values" is almost misleading because it silently changes one of the core aspects of SHAP. As an example of the consequences of this limitation, any feature that is equal to its reference value will have a SHAP value equal to zero, but it is easy to image how such features can be informative (e.g., black regions in an MNIST digit, such as missing arcs on the left-hand side of a "3" that distinguish it from an "8").

While it is very helpful to calculate SHAP values faster, this is a flawed version of SHAP that is not supported by existing research that considers the question of how to model missing features. Unfortunately, the proposed approach apparently lacks the flexibility to work with different notions of missingness (e.g., the interventional or observational approach). For that reason, I'm lowering my score by one point (6).

[1] Frye et al., "Shapley-based explainability on the data manifold" (2020)

[2] Janzing et al., "Feature relevance quantification in explainable AI: A causal problem" (2019)

[3] Sundararajan and Najmi, "The many Shapley values for model explanation" (2019)

---

> ### Author Response · Authors · 2020-11-17
> **We thank the reviewer for the positive rating, acknowledgement of our novelty, and raising valid concerns.**
>
> We will address each of them in the following according to the order in which they were raised. Note that some points are combined into one s.t. we can address them all together.
>
> 1. reference values: We chose the Baseline Shapley value method (as in [2]), as it is the most computationally efficient and simple to implement and compute for our purpose, while we believe studying how people should operationalize absence in Shapley values in explainable AI setting is of great importance.
> 2. Shallow ShapNets:
>     1. Yes, in the event that a Shallow ShapNet is used and explanations are not needed, it is equivalent to just compute the underlying function.
>     2. The reason for fast computation, as stated in the manuscript, is that we manually enforce the sparsity inside the models, which curbs the exponential complexity.
>     3. A Shallow ShapNet is acquired by summation of Shapley values for underlying functions. These intermediate values are Shapley values, so no, not any function whose output is the linear combination of intermediate values $f_j(\mathbf{x})$ be called a shallow ShapNet. Those intermediate values need to be Shapley values.
> 3. We admit that we are a bit confused by this point and we will try the best we can to respond to it (please do correct us if we are misunderstanding the question).
>     We would like to point out that we allow arbitrarily complex functions inside Shapley Modules, be it linear, non-linear, or a neural network, including those with softmax operation inside. If, however, the reviewer is wondering about the applicability to the last layer, we argue that it can still be fed into a Shapley Module as the underlying function.
> 4. We thank the reviewer for pointing out the possibility of learning the active sets (interactive pairs), but to the best of our knowledge, this kind of learning is not differentiable, making heuristic search the most likely approach, which is slow without prior by the NFL for Optimization [1]. However, these are initial thoughts, and it would indeed be interesting to investigate learning the active sets in future work (though infeasible for the current submission).
> 5.  We thank the reviewer for the suggestion and hence have moved Table 4 into the main manuscript accordingly.
>
> On the presentation problems:
> 1. Done! We thank the reviewer for pointing this out.
> 2. We have simplified Fig. 2 by reducing the number of input features from 4 to 3 for both the Shapley Module and Shallow ShapNet, which significantly reduces the number of edges present in the figure.
> 3. We have revamped the notations and the order in which Shapley Modules, Shallow/Deep ShapNets are presented.  In general, we first introduce Shallow ShapNet, which computes Shapley values for each scalar of the vector-valued function output of the underlying function $f$, followed by introducing Shapley Module as a way to limit the computational overhead, and last stack the Shallow ShapNets to form the Deep ShapNet. This should eliminate the need for a definition of generalized Shapley transform, take care of the issues with the summation operator, and the presentation is hopefully more straightforward.
> 4. baselines: We have added the baselines and sota.
> 5. Fig. 3: We thank the reviewer to bring this up and apologize for the ambiguity: this is done on MNIST dataset with Deep ShapNet averaging over the entire test set. We have updated the manuscript accordingly. Since MNIST has a dimensionality of 784, it is infeasible to compute with KernelSHAP for exact values, a sampling algorithm that requires exponential time complexity in feature dimensions, which makes large-scale sampling also an issue.
> 6. Spacing of regularization: We thank the reviewer for this suggestion and have moved one of the figures to the appendix whilst added justification for the usage of layer-wise explanations. (See below)
> 7.  For layer-wise explanation, we provide two use-cases (updated in manuscript):
>      1. we can connect the layer-wise feature importance with the layer-wise feature visualization, i.e., enables interpretation and regularization on raw features in the input space and also high-level feature in the latent space.  2. we can check if, during the entire process, a particular feature exhibits abnormally large importance values. 3. Instance-based pruning: since we know the importance of a feature, and by the missingness property, if a feature's importance is sufficiently close to zeros, we can assume the output of the next Shapley Transform will be zeros for that feature. Moreover, we can combine $\ell_1$ regularization to promote sparsity in the inter-layer representations s.t. we can prune even more dynamically during inference instead of a static pruned model.
>
> [1] D. H. Wolpert and W. G. Macready, "No free lunch theorems for optimization," in IEEE Transactions on Evolutionary Computation,
> [2] Mukund Sundararajan and Amir Najmi. The many shapley values for model explanation

---

> > ### Author Response · Authors · 2020-11-24
> > **Updated additional discussion on learning the sparsity pattern**
> >
> > We first thank the reviewer for raising thought-provoking questions, and here we discuss how we can learn the sparsity pattern induced by Shapley Module:
> >
> > As discussed with Reviewer 4, we can perform dynamic pruning during inference time, courtesy of the missingness property of Shapley values. Interestingly, this also leads to the possibility of learning the sparsity pattern: we can prune the model during training time as well, based on the importance values of each feature on the entire training dataset. To perform learning, we simply create redundancies at the beginning and slowly prune away those modules that can be pruned, which is determined by how close their input is to their reference values (since if they are close, the output is zeros by missingness).
> >
> >
> > We have added this discussion in Appendix D, and pasted here:
> >
> > >To learn the interactions, we first create redundancies in each Shallow SHAPNET inside a Deep SHAPNET.   During training,  we automatically gain the importance values for each feature by construction, from which we will determine the modules that are worth computing by comparing input representation of the features in their active sets to0s.  If the input representation is close enough to0s, we can simply ignore that module.  Of course, the underlying function can always change during training, and we argue that we can slowly strip away modules from the latter layers to the first layers as the training progresses to allow for more expressiveness and more room to correct early mistakes by the training procedure.

---

### Author Response · Authors · 2020-11-17
**We have updated our manuscript according to suggestions and concerns**

Because of the feedback that the initial submission has confusing points, we have revamped the notations and the order in which Shapley Modules, Shallow/Deep ShapNets are presented. The idea stays unchanged. We first use a tensor-based definition of Shapley representation, which allows us to better describe different dimensions of the representation and hopefully can make it cleaner. For the models, we first introduce Shallow ShapNet, which computes Shapley values for some underlying function $f$, followed by introducing Shapley Module as a way to limit the computational overhead, and lastly stack the Shallow ShapNets to form the Deep ShapNet.

We also added a subsection 2.5 in which we discuss the implications of layer-wise explanation, which mainly focuses on dynamic conditional pruning, enabled by a simple extension of missingness property in Deep ShapNet: if a feature has importance values close to zeros at any stage, it will always be zeros in the following stages.


We also added a few experiments (though not all) requested by the reviewers, and we thank the reviewers for the improvement that we were able to make because of their constructive feedback.

---

### Decision · Program_Chairs · 2021-01-07
**Final Decision**

**Decision:**

Accept (Poster)

**Comment:**

Shapley values are an important approach in extracting meaning from trained deep neural networks, and the paper proposes an innovative approach to address inefficiencies in post-processing to compute Shapley values, by instead incorporating their computation into training.  There was a robust discussion of this paper, and the authors' comments and changes substantially strengthened the paper and the reviewers' view of it, to the point that all reviewers now recommend acceptance.  Some lingering concerns remain that the authors should continue to work to address.  Is the method of computing Shapley values used as the baseline in the paper really state-of-the-art, or artificially weak?  The empirical results were methodologically sound but not as strong as one might expect or hope.  These concerns detract somewhat from enthusiasm, but nevertheless the paper yields an innovation to a widely-used approach to one of the most pressing current research problems.  The reviewers had a number of smaller suggestions that should also be incorporated including more significance testing and reporting of resulting p-values.